# Fasciclin 2 functions as an expression-level switch on EGFR to control organ shape and size in *Drosophila*

Luis Garcia-Alonso *

Instituto de Neurociencias CSIC-UMH, Universidad Miguel Hernandez, Sant Joan d'Alacant, Alicante, Spain

* lgalonso@umh.es

## Abstract

Fasciclin 2 (*Drosophila* NCAM) is a homophilic Cell Adhesion Molecule expressed at moderate levels in the proliferating epithelial cells of imaginal discs, where it engages EGFR in a cell autonomous auto-stimulatory loop that promotes growth along larval development. In addition, Fasciclin 2 is expressed at high levels in the pre-differentiating cells of imaginal discs. Gain-of-function genetic analysis shows that Fasciclin 2 acts as a non-cell autonomous repressor of EGFR when high expression levels are induced during imaginal disc growth. Loss-of-function genetic analysis shows that this Fasciclin 2 functional facet is required at the end of larval development and it is mediated by interaction with IgCAMs CG15630 (Fipi) and CG33543 (Elff). Thus, Fasciclin 2 bears two complementary functional roles which correspond with different levels of expression. The combined results from loss- and gain-of-function analyses suggest a scenario where the Fasciclin 2/EGFR cell autonomous auto-stimulatory loop promotes cell proliferation until reaching a Fasciclin 2 expression threshold where its non-cell autonomous function stops growth. Thus, cellular integration of Fasciclin 2 autonomous and non-cell autonomous signaling from neighbor cells may be a key regulator component to orchestrate the rate of intercalary cell proliferation and the final size and shape of an organ.

## Introduction

During organ morphogenesis growth is tightly controlled to ensure the species-specific constancy of pattern, size and shape. How organs control the termination of local cell proliferation when they reach a species-specific shape and size is a central question in developmental biology. Although the mechanisms involved in this process are not well understood, it is clear that they occur at different organization levels, from systemic (see [1]) and tissue (see [2]) to local cell interactions (see [3]). Local control of intercalary growth is critical to ensure the constancy of proportions, the distance between pattern elements and the shape of an organ. It is still an unsolved problem how cells recognize when local growth has reached a species-specific amount to stop proliferation. Cell to cell contact interactions can provide the high level of precision required to compute the amount of local tissue growth [3]. Moreover, the coincident

**Funding:** This work was supported by PID2021-123407OB-I00, CEX2021-001165-S and BFU2016-

76295-R from MCIN/AEI/10.13039/501100011033 and by "ERDF a way of making Europe"; PROMETEU 2021-027 from Generalitat Valenciana, and SAF2004-06593 from MCYT. The funders had no role in study design, data collection and analysis, decision to publish, or preparation of the manuscript.

action of cell-cell contact recognition and diffusible growth factors on specific receptors may generate the dynamics and the level of precision required for the fine control of local growth (see [4]). Immunoglobulin superfamily cell adhesion molecules (IgCAMs) of the L1-CAM- and NCAM-type families mediate highly specific cell recognition and control of Receptor Tyrosine Kinase signaling during axon growth and guidance [5,6].

Fasciclin 2 (Fas2, *Drosophila* NCAM) is a homophilic IgCAM dynamically expressed by all epithelial cells in imaginal discs, with maximal peaks of expression in pre-differentiating and differentiating structures [7,8]. Fas2 can also participate in heterophilic binding with other specific IgCAMs through extracellular domain interactions [9]. Our previous results have shown that Fas2 and EGFR engage in a cell autonomous auto-stimulatory loop that promotes EGFR activity and Fas2 expression during imaginal disc growth [8]. Conversely, loss-of function (LOF) analysis has shown that Fas2 functions as a non-cell autonomous repressor of EGFR during retinal differentiation [10]. Here, the study of Fas2 in gain-of-function (GOF) conditions using coupled-MARCM [11] and FLP-OUT genetic mosaics shows that Fas2 can inhibit cell proliferation by a non-cell autonomous local repression of EGFR during imaginal disc growth. This function requires the interaction of Fas2 with its heterophilic binding partners CG15630 (*fipi*, [12]) and CG33543 (referred in the rest of this work as *elff*, for *epithelial limiter of fas2 function*). Both Fipi and Elff IgCAMs are expressed in overlapping proximity to Fas2 in the basolateral membrane of epithelial cells during imaginal disc growth. Inhibition of *fipi* and *elff* expression suppresses the Fas2 non-cell autonomous GOF phenotype.

The combined results from LOF and GOF analyses suggest a scenario where the expression of Fas2 would operate as a level switch for growth. Fas2 interaction with the EGFR could promote growth until reaching a Fas2 expression threshold where the interaction with Fipi and Elff would stop local growth in a non-cell autonomous manner. Consistent with this scenario, a mild inhibition of Fas2 expression just before metamorphosis is sufficient to cause extra-growth. Moreover, simultaneous high levels of both Fas2 and the diffusible EGFR activator ligand Vein synergize coupling repression of cell proliferation and wing vein differentiation.

## Results

### Fas2 over-expression induces an EGFR-dependent non-cell autonomous restrain of growth during imaginal disc development

The trans-membrane and GPI-linked isoforms of Fas2 (Fas2$^{TRM}$ and Fas2$^{GPI}$) are expressed in epithelial cells during larval development [13,14]. Both native isoforms can activate EGFR and rescue the *fas2* null condition [8]. Over-expression of either isoform of Fas2 (Fas2$^{GOF}$) driven by heterozygous *MS1096-GAL4/+* in the wing imaginal disc caused a dose-dependent reduction in the size of the adult wing (Fig 1A and 1C). Generalized over-expression of Fas2 driven by heterozygous *Tubulin-GAL4/+* caused a complete body size reduction compared with control siblings of the same sex (S1A Fig). Compartment-specific over-expression of Fas2 with the heterozygous *engrailed-GAL4/+* driver caused an abnormal wing shape due to the reduction in size of the posterior compartment (Fig 1B) without affecting cell size (S1B Fig). In these wings, the Fas2$^{GOF}$ condition in the posterior compartment also reduced the size of the adjacent region in the anterior compartment (between vein III and the A/P compartment border), but not the most distant one (between vein III and the anterior border of the wing; Fig 1B). Fas2 over-expression in *eyeless*-driven FLP-OUTs caused a size reduction of the head capsule (Fig 2A).

FLP-OUT Fas2 over-expression clones could cover a very significant fraction of the adult notum but caused a striking reduction in the size of the organ (Fig 1D–1F). This suggests a non-cell autonomous effect of the clones on the normal territory, which could not compensate

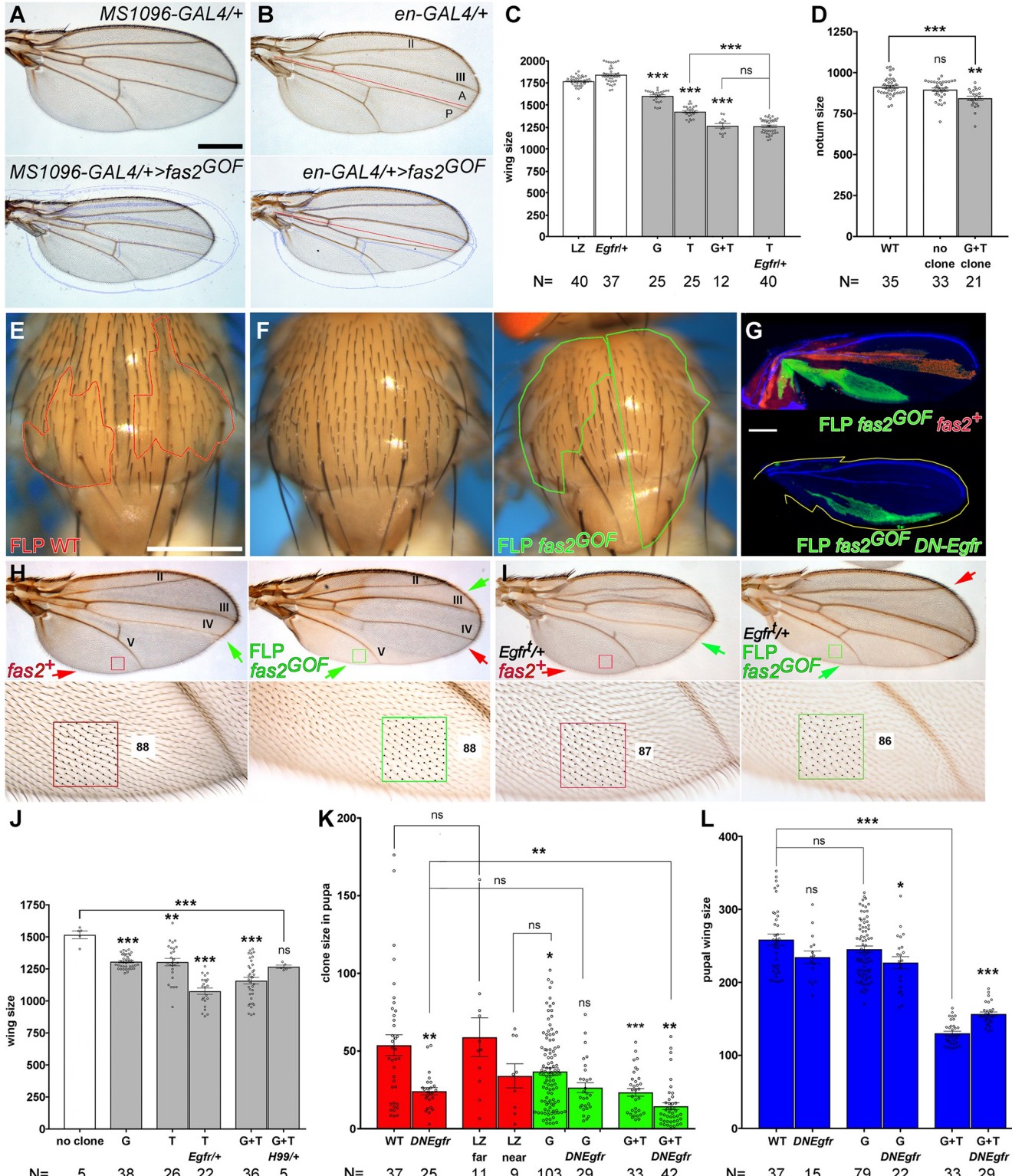

**Fig 1. Over-expression of Fas2 in the wing imaginal disc causes an EGFR-dependent reduction in organ size.** (A) Top, control *MS1096-GAL4/+* driver. Bottom, over-expression of Fas2 (*fas2^GOF*) in *MS1096-GAL4/+; UAS-fas2^TRM UAS-fas2^GPI/+* heterozygous females causes a general reduction of wing size. The background blue outline corresponds to the control wing on top. Bar: 500 μm. (B) Over-expression of Fas2 (*fas2^GOF*) in *en-GAL4/UAS-fas2^TRM UAS-fas2^GPI* females causes a size reduction in the posterior compartment (bottom). The effect is local but extends into the neighbor region of the anterior compartment, close to the compartment border (red line). Note also the perfect match in the most anterior region of the wing with the blue outline of the

control wing on top. (C) *MS1096-GAL4* driven over-expression of Fas2 causes a dose-dependent wing size reduction. This phenotype is significantly enhanced by the heterozygosity for the *Egfr* mutant *top* in *MS1096-GAL4/+; UAS-fas2*<sup>TRM</sup>/Egfr<sup>top</sup> individuals (T *Egfr*/+). This enhanced phenotype becomes similar to that obtained by the simultaneous over-expression of Fas2<sup>GPI</sup> plus Fas2<sup>TRM</sup> (G+T). All individuals were heterozygous females. Wing size is area in μm²/10³. (D) The notum size of adult male flies bearing FLP-OUT clones over-expressing Fas2<sup>GPI</sup> and Fas2<sup>TRM</sup> (G+T clone) was significantly smaller than the nota from either WT or siblings without clones (WT, no clone, white bars). Note that notum size area is a relative value due to the curvature of the notum. (E) FLP-OUT notum control clones in males (*y w hs-FLP; ActinFRTy*⁺*FRT-GAL4 UAS-GFP/+*) induced in 1ˢᵗ instar larva (*yellow* marker, outlined in red). Bar: 500 μm. (F) FLP-OUT notum clones over-expressing Fas2 (*fas2*<sup>GOF</sup>) in *y w hs-FLP; ActinFRTy*⁺*FRT-GAL4 UAS-GFP/UAS-fas2*<sup>TRM</sup> *UAS-fas2*<sup>GPI</sup> males induced in 1ˢᵗ instar larva. At left, control sibling without clones. At right, individual bearing two clones (*yellow* cuticle, outlined in green), note the large size of the clones and the general reduction of notum size. All studied individuals were male. (G) Top, a FLP-OUT *fas2*⁺ (LacZ, red) internal control clone in the anterior compartment and a *fas2*<sup>GOF</sup> (GFP, green) clone in the posterior compartment in a *y w hs-FLP; ActinFRTy*⁺*FRT-GAL4 UAS-GFP/UAS-fas2*<sup>GPI</sup>; *ActinFRTpolyAFRT-LacZ/+* pupal wing. Bottom, a *fas2*<sup>GOF</sup> *DN-Egfr* clone in the posterior compartment of a *y w hs-FLP; ActinFRTy*⁺*FRT-GAL4 UAS-GFP/UAS-fas2*<sup>GPI</sup>; *UAS-DN-EGFR/+* pupal wing. The yellow outline in the *fas2*<sup>GOF</sup> *DN-Egfr* clone corresponds with the wing above (without *DN-Egfr*). Clones induced in 1ˢᵗ instar larva. Bar: 50μm. (H) Top, two adult *y w hs-FLP; ActinFRTy*⁺*FRT-GAL4 UAS-GFP/UAS-fas2*<sup>TRM</sup> male wings bearing FLP-OUT clones (*yellow* marker) over-expressing Fas2<sup>TRM</sup> (*fas2*<sup>GOF</sup>) in different regions. At left, a clone in the Anterior compartment between veins III and IV (green arrow) causes an abnormal shape due to a reduction of size in the intervein region. At right, a clone in the Posterior compartment between vein V and the posterior wing margin (green arrow) causes an abnormal shape due to the reduction in size of the posterior region of the wing, while another clone between veins II and III causes a size decrease in the anterior compartment (green arrow). The red arrows point to normal territory. Bottom, magnification of the areas in the frames on top. Left, normal region (red square) in the wing on top to compare with the same region in the wing at right covered by a Fas2<sup>TRM</sup> GOF clone (green square). The *fas2*<sup>GOF</sup> area at right (green square) shows the exact same number of trichomes (each one corresponding to a single cell) than the *fas2*⁺ territory at left (red square), ruling out putative changes in cell size caused by the Fas2 GOF condition. Note the reduction in cell number between vein V and the wing margin caused by the Fas2<sup>GOF</sup> clone at right. (I) Two *y w hs-FLP; ActinFRTy*⁺*FRT-GAL4 UAS-GFP/UAS-fas2*<sup>TRM</sup> *Egfr*<sup>top</sup> male wings with FLP-OUT clones over-expressing Fas2<sup>TRM</sup> in a genetic background with a 50% reduction of *Egfr* dosage (*Egfr*<sup>top</sup>/+ FLP *fas2*<sup>GOF</sup>, green arrows). Both wings have clones in positions similar to those in Fig 1H. The wing at the left has a clone causing a strong reduction of size between veins III and IV, but it is normal from vein V to the posterior wing margin (red arrow). The wing in the right picture has a clone covering most of the Posterior compartment, causing a strong size reduction of the posterior wing but not of the most anterior wing (red arrow). The *Egfr* heterozygous condition of these wings enhances the Fas2 over-expression phenotype (compare with Fig 1H; quantified in Fig 1J). Moreover, this enhancement is due to a lower number of cells, since the number of cells (trichomes) is unchanged compared to WT territory (compare red and green squares, and also with Fig 1H). Clones are marked with *yellow*. (J) Quantitative analysis of size in adult wings bearing Fas2 GOF clones. Wings bearing FLP-OUT clones over-expressing Fas2<sup>GPI</sup> (G), Fas2<sup>TRM</sup> (T), Fas2<sup>TRM</sup> in an *Egfr*<sup>top</sup>/+ background (T *Egfr*/+); Fas2<sup>GPI</sup> plus Fas2<sup>TRM</sup> (G+T) and Fas2<sup>GPI</sup> plus Fas2<sup>TRM</sup> in a *Df(3L)H99/+* genetic background (G+T *H99/+*) are compared with wings from siblings without clones (no clone). FLP-OUT clones expressing one *fas2* GOF single insertion caused a significant reduction in adult wing size, while clones expressing two *fas2* GOF insertions displayed a more pronounced reduction of wing size. The phenotype is caused by a repression of EGFR function, as revealed by its enhancement in an *Egfr*<sup>top</sup>/+ genetic background. Wing size is wing area in μm²/10³. All individuals were males. (K) Quantitative size analysis of Fas2 GOF clones in pupal wings. The size of FLP-OUT clones over-expressing Fas2<sup>GPI</sup> (G) or Fas2<sup>GPI</sup> plus Fas2<sup>TRM</sup> (G+T) (green bars) is smaller than the size of FLP-OUT WT clones (WT, red bar) or LacZ (LZ far, red bar) internal controls growing apart from a Fas2<sup>GPI</sup> neighbor clone. Clones over-expressing Fas2<sup>GPI</sup> (G) and LacZ internal control clones in the same compartment (LZ near, red bar) were of similar size. The size of pupal FLP-OUT clones over-expressing Fas2<sup>GPI</sup> and a dominant negative form of EGFR (G *DNEgfr*) is not significantly different from clones expressing DN-EGFR (*DNEgfr*) or Fas2<sup>GPI</sup> (G) alone. The combination expressing Fas2<sup>GPI</sup> plus Fas2<sup>TRM</sup> is enhanced by the presence of DN-EGFR (G+T *DNEgfr*). Clone size is area in μm²/10³. (L) The non-cell autonomous effect of the Fas2<sup>GOF</sup> clones (in K) on the size of pupal wings is not normalized by the cell autonomous expression of DN-EGFR. Pupal wing size is area in μm²/10³.

for the loss of growth to attain a normal size. Adult wings with Fas2 over-expression clones displayed strong shape alterations associated with a reduction in the number of cells in the region bearing the clone (Fig 1H, top panels). In these clones, spacing and polarity of trichomes (which mark each single epidermal cell) was normal (Fig 1H, bottom panels). The size of wings bearing these Fas2<sup>GOF</sup> clones was reduced in a *fas2* dose-dependent manner (Fig 1J). To quantify the capacity of Fas2 over-expression to occupy imaginal disc territory and produce a consistent similar number of clones in each imaginal disc of Fas2 GOF and WT controls, I studied *ey>FLP*-driven MARCM clones in the eye disc (Fig 2B). These clones caused a reduction of eye disc size compared to WT clone controls as expected (Fig 2C). Remarkably, these clones covered a significant larger fraction of the imaginal disc than WT controls did (Fig 2D), suggesting that they initially grow larger than normal but stop growth earlier. In pupae, wing discs bearing FLP-OUT Fas2<sup>GOF</sup> clones (Fig 1G, top panel) displayed a dose dependent reduction of size and caused a decrease of the whole wing size (Fig 1K and 1L). Interestingly, endogenous FLP-OUT control clones growing close (same compartment) to the Fas2 GOF clones displayed a similar reduced size, while control clones growing far (different compartment) were similar to WT (Fig 1K). Fas2 GOF clones growing in the wing imaginal disc did not show signs of apoptosis by either expression of cleaved-Caspase3 or expression of the *puc-LacZ* reporter [15] (S1C Fig). Moreover, the effect of Fas2<sup>GOF</sup> clones on wing size was not eliminated

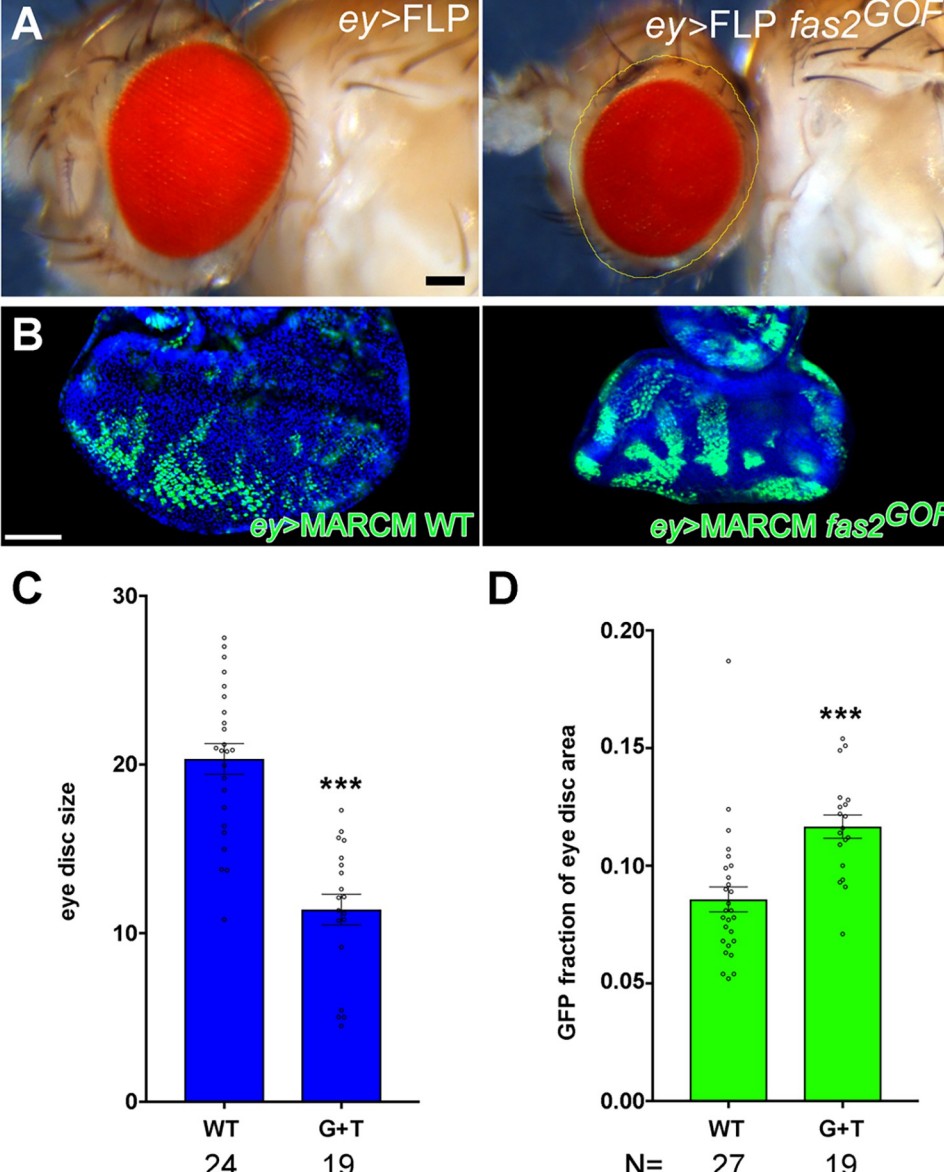

**Fig 2. The Fas2 GOF condition in eye disc clones allows them to colonize a larger fraction of the disc than normal cells but causes a general reduction of organ size.** (A) Over-expression of Fas2 (*fas2^GOF^*) in a male *y ey-FLP; ActinFRTy⁺FRT-GAL4 UAS-GFP/UAS-fas2^TRM^ UAS-fas2^GPI^* FLP-OUT causes a reduction in head size. Outline in yellow (bottom) corresponds to eye size in the control (*CyO* balancer sibling) on top. (B) A consistent similar number of MARCM clones in each eye imaginal disc can be produced by *ey-flp* driven mitotic recombination for WT controls (left) and Fas2 GOF (Fas2^TRM^ plus Fas2^GPI^, right). Bar: 50 μm. (C) Eye imaginal discs with *ey-FLP*-driven MARCM Fas2^GOF^ clones (*UAS-fas2^GPI^ UAS-fas2^TRM^*/+, G+T) display a strong size reduction compared to WT controls. (D) *ey-FLP*-driven MARCM Fas2^GOF^ clones (*UAS-fas2^GPI^ UAS-fas2^TRM^*/+, G+T) cover a significantly larger area of the eye disc than WT controls.

in *Df(3L)H99/+* individuals, which have a general suppression of apoptosis [16] (Fig 1J). Thus, Fas2^GOF^ cells actively restrain the growth of their neighbors by a mechanism not involving apoptosis or JNK signaling.

Conditions causing either a reduction [17] or a gain in *Egfr* signaling [18,19] can cause a decrease in wing size. To analyze if the Fas2 GOF phenotype results from a gain or a loss of

EGFR activity, I combined a reduction in half of the *Egfr* dosage with the over-expression condition for Fas2$^{TRM}$. This combination showed a significant enhancement of the phenotype, which became as strong as that produced by the over-expression of Fas2$^{GPI}$ and Fas2$^{TRM}$ together (Fig 1C). A similar *Egfr* dosage decrease in wings bearing Fas2 over-expression clones caused an enhancement of the phenotype without affecting the spacing or polarity of trichomes (Fig 1I, compare with Fig 1H; quantified in Fig 1J). Fas2$^{GOF}$ FLP-OUT clones expressing constitutively activated-EGFR *(λEgfr)* showed the same phenotype than Fas2$^{GOF}$ clones simultaneously expressing activated-EGFR (S1D Fig), consistent with the epistatic relationship between Fas2 and EGFR functions [8]. Thus, the genetic interactions of Fas2 over-expression with *Egfr* LOF and GOF conditions indicate that a repression of EGFR activity mediates the Fas2 GOF phenotype.

Since Fas2 promotes cell autonomous EGFR activity during imaginal disc growth [8], the observed non-cell autonomous phenotype of the Fas2 over-expression may had simply reflected the release of some diffusible feedback repressor downstream of EGFR from the clones with a high EGFR activity (see as an example [20,21]). However, if this had been the case, we would have had to expect the wings bearing the Fas2 GOF clones developing in the *Egfr*/+ heterozygous background to be normalized compared to those with Fas2 GOF clones developing in an *Egfr*$^+$ normal background (due to the corresponding reduction in the liberation of the putative feedback repressor). As shown before, this was not the case (Fig 1J). Moreover, expression of DN-EGFR in Fas2 over-expression clones showed the expected epistasis of *Egfr* over *fas2* within the clones (note that EGFR activity is cell autonomous and mediates the Fas2 function inside the clones [8]) (Fig 1G, bottom panel; Fig 1K), but the pupal wings bearing these clones did not show any normalization of the non-cell autonomous effect of these Fas2 GOF clones on wing size (Fig 1G and 1L). The results are strongly consistent with a direct non-cell autonomous repression of EGFR by the Fas2 over-expression during imaginal disc growth.

To further characterize the non-cell autonomous component of the Fas2 GOF phenotype, I quantified the number of mitoses in FLP-OUT clones and the size of control twin clones in coupled-MARCM mosaics. Control FLP-OUT WT clones and the rim of 2–3 cells around them displayed the same mitotic index (Fig 3A and 3B), but the mitotic index in FLP-OUT Fas2$^{GOF}$ clones and the rim of 2–3 normal cells around them was significantly reduced. Coupled-MARCM Fas2$^{GOF}$ clones in the wing disc were smaller than their control twins (Fig 3C and 3D). In turn, these control twin clones were significantly smaller than WT controls (Fig 3D), while the size of the whole wing disc was reduced compared with WT (Fig 3D and 3E). Thus, the data indicates an instructive non-cell autonomous effect of the Fas2$^{GOF}$ clone on the normal cells around the clone.

The results together are strongly consistent with the Fas2 GOF condition initially causing a cell autonomous enhanced growth but then repressing it in a non-cell autonomous manner before reaching the species-specific final size of the imaginal disc.

## Fas2 limits imaginal disc growth before metamorphosis

Fas2 promotes cell autonomous EGFR-dependent growth during imaginal disc development [8], while it represses EGFR during retinal differentiation [10] and in over-expression conditions during imaginal disc growth (results above). Do these two functional facets of Fas2 represent alternative and independent forms of action in different developmental scenarios (proliferative vs. differentiating tissue)? or, do they integrate in a functional logic during development? Indeed, the positive feedback between Fas2 and EGFR during imaginal disc growth and its higher expression in pre-differentiating imaginal disc structures [7,8] suggest that Fas2

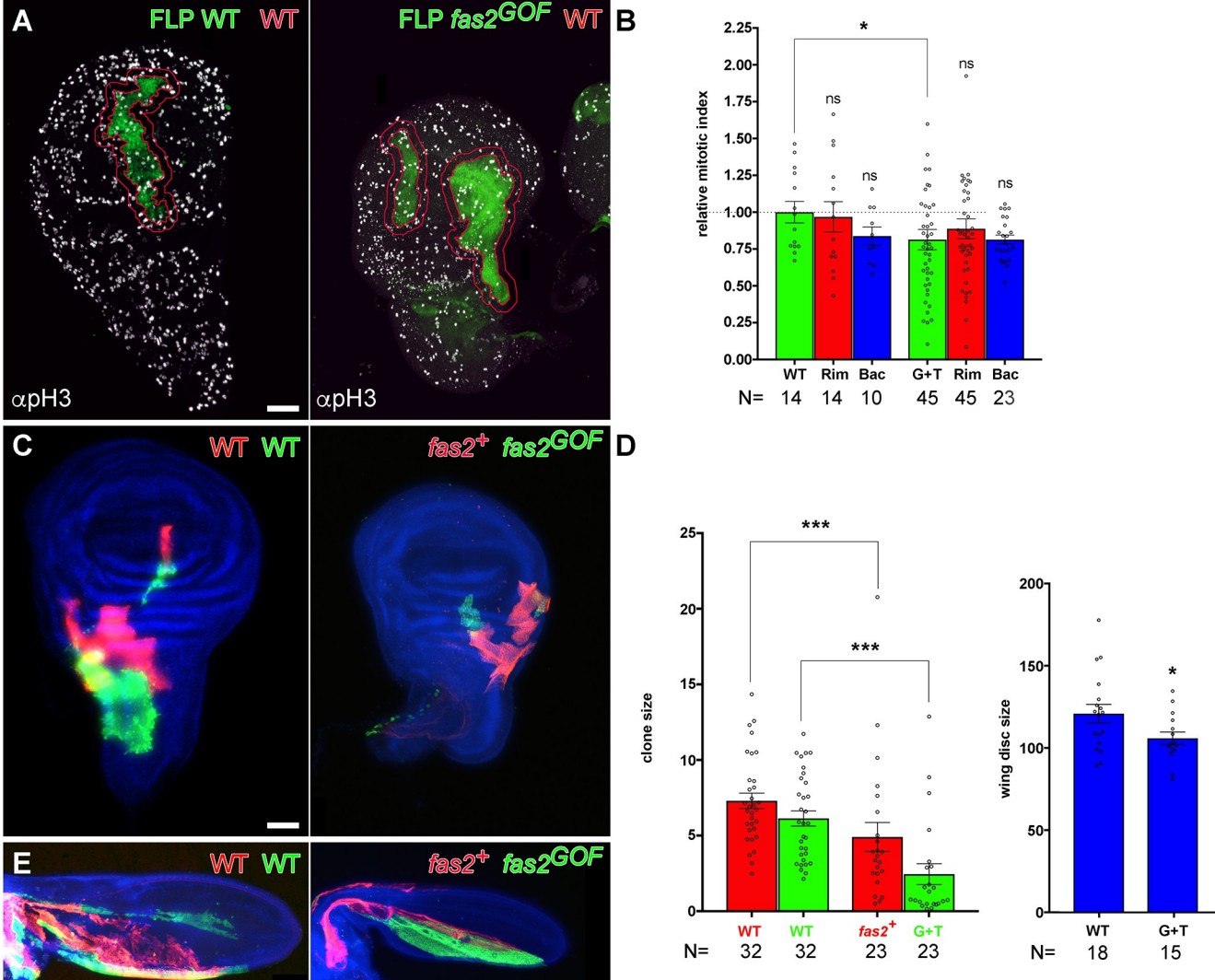

**Fig 3. Over-expression of Fas2 causes a non-cell autonomous restraining of growth.** (A) FLP-OUT clones expressing Fas2$^{GPI}$ plus Fas2$^{TRM}$ (fas2$^{GOF}$) in y w hs-FLP; ActinFRTy$^+$FRT-GAL4 UAS-GFP/UAS-fas2$^{TRM}$ UAS-fas2$^{GPI}$ imaginal discs show a reduced number of mitosis (anti-pH3 antibody, right picture) compared with control WT FLP-OUT clones in y w hs-FLP; ActinFRTy$^+$FRT-GAL4 UAS-GFP/+ imaginal discs (left picture). An outline (red rim) of 2–3 cell diameters of normal tissue around the perimeter of the clones was used to quantify the number of mitosis in the clone, the rim and the rest of the imaginal disc. (B) Quantification of mitotic index in the Fas2$^{GOF}$ FLP-OUT clones and in their normal neighbor cells. Clones expressing Fas2$^{GPI}$ and Fas2$^{TRM}$ (G+T, green bar) displayed a significant reduction in the number of mitoses compared with WT clones (WT, green bar). Paired t-test comparison between each clone (G+T) and the rim of 2–3 normal cells around it (RIM, red bar) showed no difference in mitotic index. The background mitotic index in the rest of the disc (Bac, blue bar) was similar to WT. Y-axis, number of mitosis/$\mu m^2$ relative to w$^{1118}$ control (WT) FLP-OUT clones. (C) Coupled-MARCM analysis of the Fas2$^{GOF}$ condition in w hs-FLP Tub-GAL80 FRT19A/y QS13F FRT19A; UAS-fas2$^{GPI}$ UAS-fas2$^{TRM}$/QF-ET40 QUAS-mtdTomato; Tub-GAL4 UAS-GFP/+ and WT (w hs-FLP Tub-GAL80 FRT19A/y QS13F FRT19A; QF-ET40 QUAS-mtdTomato/+; Tub-GAL4 UAS-GFP/+) wing imaginal discs. Cell clones over-expressing Fas2$^{GPI}$ and Fas2$^{TRM}$ (fas2$^{GOF}$, GFP) were smaller than their control normal twins (fas2$^+$, mtdTomato). Clones induced in 1$^{st}$ instar larva. Bar: 50 μm. (D) Left, quantitative analysis of coupled-MARCM twin clones in the wing disc. Fas2$^{GOF}$ clones (G+T, UAS-fas2$^{GPI}$ UAS-fas2$^{TRM}$/+, green bar) were smaller than their control twins (fas2$^+$, red bar). In addition, both types of twin clones were significantly smaller than WT coupled-MARCM control clones, showing the presence of a non-cell autonomous restraining of growth caused by the Fas2$^{GOF}$ clones. Right, wing imaginal discs bearing coupled-MARCM Fas2$^{GOF}$ clones had a significantly smaller size than the WT wing disc controls. Clone size and wing disc size expressed as area in $\mu m^2/10^3$. Clones induced in 1$^{st}$ instar larva. (E) Coupled-MARCM analysis of the Fas2$^{GOF}$ condition in pupal wings. Left, control WT twin clones. Right, over-expression of Fas2$^{GPI}$ plus Fas2$^{TRM}$ (fas2$^{GOF}$, UAS-fas2$^{GPI}$ UAS-fas2$^{TRM}$/+) in clones caused a reduction in pupal wing size. Clones induced in 1$^{st}$ instar larva.

expression may increase to reach a threshold where its non-cell autonomous component could restrain cell proliferation. The existence of an EGFR positive feedback amplification loop required for vein formation in wing imaginal discs has been proposed before [22], as well as the existence of a Fas2 expression threshold for synapse growth and function [23,24]. During synapse development the maximum growth is attained at 50% of normal Fas2 expression, while larger or smaller amounts cause synapse size and function reductions. The data presented above suggested that a threshold for Fas2 expression to switch from EGFR activator to EGFR repressor may exist in imaginal discs. Indeed, a reduction of 50% in *fas2* dosage (*fas2*$^{eB112}$/+) causes larger wings (Fig 4A). If this inference holds true, a mild inhibition of Fas2 expression during imaginal disc development may maintain cell proliferation but also delay reaching the threshold level and cause a larger wing.

The *hs-GAL4* (RRID: BDSC_1799) driver has a leaky expression at 25˚C that can be used to produce a mild inhibition of Fas2 expression. As previously reported [8], inhibition of Fas2 expression in heterozygous individuals carrying *UAS-fas2*$^{RNAi}$ (RRID:BDSC_28990, Ria) or *UAS-fas2*$^{RNAi}$ (RRID:BDSC_34084, Rib) and the *Tub-GAL4/+* or *MS1096-GAL4/+* drivers at 25˚C caused a wing size reduction compared with controls expressing *UAS-LacZ*. In contrast, combinations of the *hs-GAL4/+* driver with either *fas2* RNAi raised at 25˚C caused a significant increase in wing size compared with the *UAS-LacZ* controls (Fig 4A), while raising the same combinations at 29C (which enhances *hs-GAL4* expression) already caused a significant wing size reduction (Fig 4A).

To test if both functional facets of Fas2 as cell autonomous activator and non-cell autonomous repressor of EGFR coexist just prior to metamorphosis, I combined *Tub-GAL80ts* with either *MS1096-GAL4/+* or *hs-GAL4/+* to specifically release the expression of *fas2*$^{RNAi}$ (or control *UAS-LacZ*) by the end of larval development. Inhibition of Fas2 expression by a shift from permissive to restrictive temperature during late third instar larva caused significant reductions of wing size with the *MS1096-GAL4/+* driver, demonstrating that Fas2 is required for growth until the end of imaginal disc growth (Fig 4B). In contrast, using the *hs-GAL4/+* driver, the mild inhibition of Fas2 by the expression of *RNAi*$^{BDSC\_34084}$ (Rib) by the end of larval development caused an increase in wing size. A similar result was obtained by giving a single 24 hours pulse at 25˚C before puparium formation to individuals continuously raised at permissive temperature (Fig 4B). Therefore, both functional facets of Fas2, as non-cell autonomous EGFR repressor and cell autonomous EGFR activator, co-exist by the end of larval life.

## Expression of Fipi and Elff IgCAMs is required for the non-cell autonomous function of Fas2

The previous results indicated that the integrated Fas2 control of EGFR activity switches from net activation to net repression at high expression levels, suggesting the involvement in this transition of a new functional interaction that engages Fas2 in high expression conditions. Since the Fas2 GPI-anchored isoform can sustain both, the cell autonomous and the non-cell autonomous functional facets ([8] and results above), an interaction between the extracellular domain of Fas2 with some other protein on the cell membrane would likely mediate its non-cell autonomous function. Interestingly, it has been shown that IgCAMs coded by the genes *CG15630* (*fipi*) and *CG33543* (referred as *elff* in this paper) can bind Fas2 through extracellular domain interactions [9]. The functional significance of these protein interactions with Fas2 is unknown.

Protein trap Fipi-GFP (RRID: BDSC_#60532) and Elff-GFP (RRID: BDSC_60531) [25] expression was detected in all epithelial cells of imaginal discs (Fig 5A and 5B). While Fas2 is expressed at the basolateral junctions in epithelia [26], these proteins occupy a closely adjacent

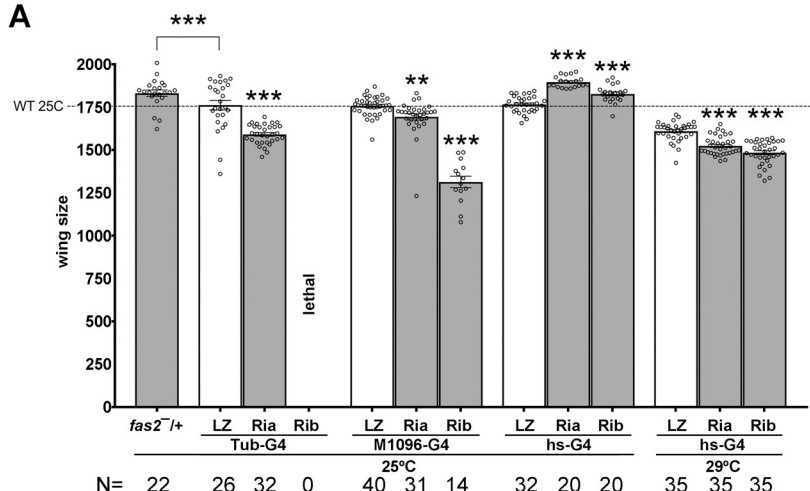

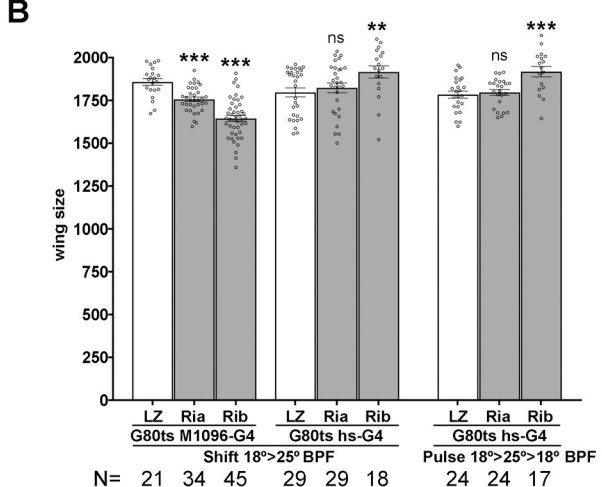

**Fig 4. Fas2 is required to limit imaginal disc growth before puparium formation.** (A) RNAi inhibition of Fas2 expression with strong *GAL4* drivers compared to inhibition produced by a weak driver. Strong inhibition caused by heterozygous *Tubulin-GAL4/+* or *MS1096-GAL4/+* at 25°C causes a restraining of wing disc growth for either *fas2 RNAi* RRID: BDSC_28990 or *RNAi* RRID: BDSC_34084 (Ria and Rib respectively) compared to controls expressing LacZ. In contrast, inhibition of *fas2* by the leaky expression of *hs-GAL4/+* (RRID:BDSC_1799) at 25°C causes extra-growth for both RNAis, similar to that obtained by reducing the *fas2* gene dosage by 50% (*fas2^eB112^/+*, *fas2^−^/+*). Increasing *hs-GAL4* expression by raising the larvae at 29°C causes these combinations to display a reduced wing size compared to controls. All female individuals. Wing size are area in $\mu m^2/10^3$. (B) Selective inhibition of Fas2 expression during late larval development. Quantification of wing size in flies raised at 18°C and either shifted to 25°C or exposed to a 25°C pulse for 24 hours during the last half of 3^rd instar development. At left (Shift 18˚>25˚ BPF), *MS1096-GAL4/+; Tub-GAL80^ts20^/+; UAS-fas2^RNAi^/+* (Ria, RRID:BDSC_28990; Rib, RRID:BDSC_34084) and *Tub-GAL80^ts20^/+; hs-GAL4/UAS-fas2^RNAi^* (Ria, RRID:BDSC_28990; Rib, RRID:BDSC_34084) individuals compared with *UAS-LacZ* controls (LZ). While the strong *MS1096-GAL4/+* heterozygous driver causes the typical restraining of growth associated with Fas2 LOF conditions, the partial mild inhibition of *fas2* by the leaky expression of *hs-GAL4* at 25°C by the end of larval life caused a significant increase in wing size for *RNAi* RRID: BDSC_34084. At right, a similar result was produced by a 25°C pulse for 24 hours during late 3^rd instar larva (Pulse 18˚>25˚>18˚ BPF). All female individuals. Wing size is area in $\mu m^2/10^3$.

and partially overlapping, but more apical position to Fas2 (see Figs 5B and S2). Remarkably, *fipi-GFP* eye imaginal discs showed a wave of high expression preceding the high wave expression of Fas2 in the morphogenetic furrow, while differentiating retinal cells showed extensive overlap between them (Fig 5A, bottom).

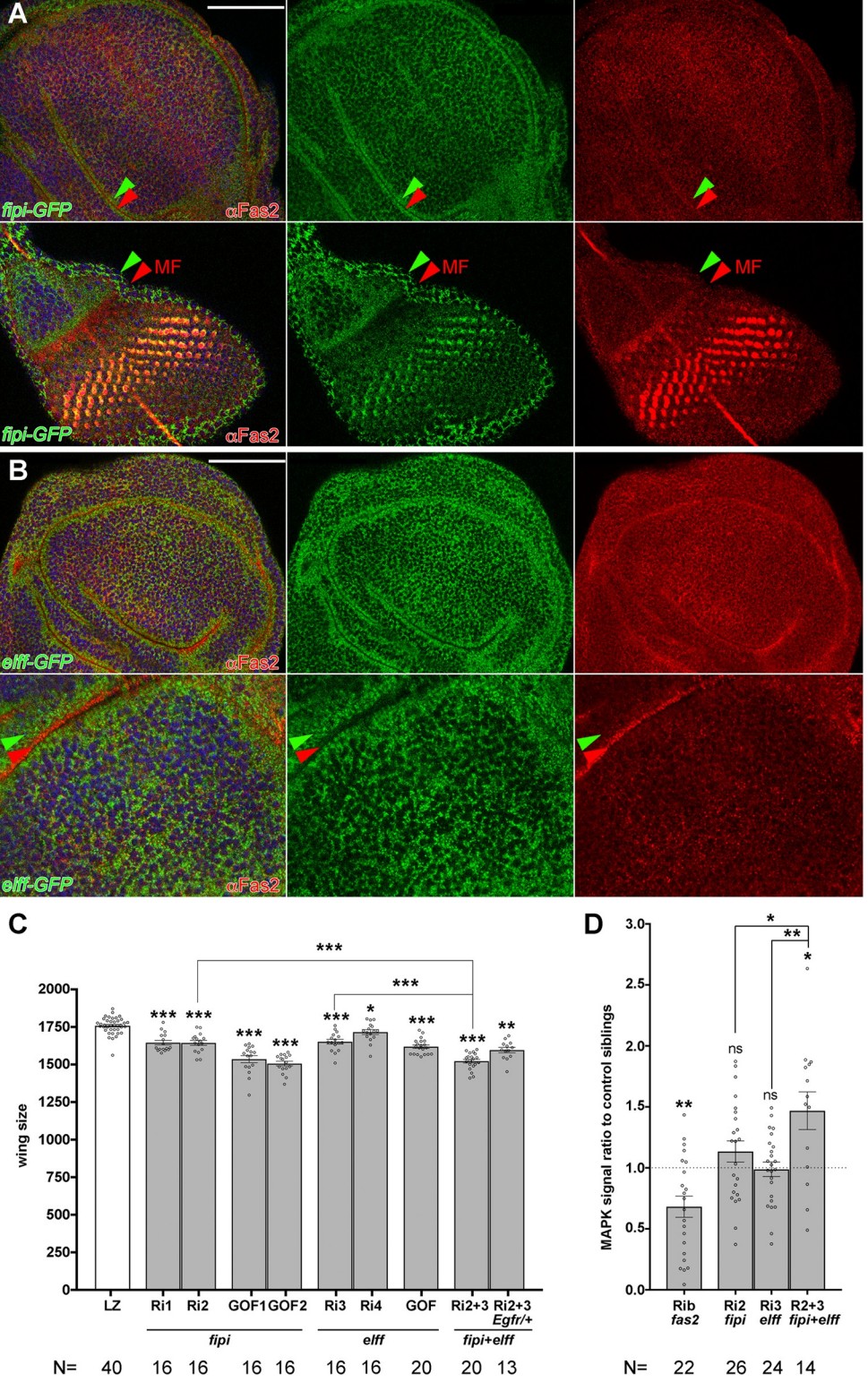

**Fig 5. Fas2 interacting partner IgCAMs Fipi and Elff are required for imaginal disc development.** (A) Expression pattern of Fipi IgCAM in imaginal discs. The protein trap CG15630-GFP-Flag RRID: BDSC_60532 is expressed in all epithelial cells of imaginal discs along with Fas2 (αFas2, 1D4 antibody). The expression of this protein does not exactly match Fas2 within the cells (see also S2A Fig for a lateral view of the cells in the discs). Note the little overlap between red (Fas2, red arrowhead) and green (Fipi protein trap, green arrowhead) in the wing disc (top) and eye imaginal disc

before the morphogenetic furrow (MF, Fas2 red arrowhead, bottom). However, there is extensive overlap with Fas2 in the differentiating retina of the eye imaginal disc after the MF, where Fas2 expression is maximal (yellow cells, bottom left picture). Bar: 50 μm. (B) Expression pattern of Elff IgCAM in imaginal discs. The protein trap CG33543-GFP-Flag RRID: BDSC_60531 shows a pattern of expression similar to Fipi relative to Fas2 (red arrowhead, bottom panels) in the wing imaginal disc. The confocal lateral view of the epithelial cells in the disc folds shows Fas2 (red) more basally located than Elff (green arrowhead; see also S2B Fig). Bar: 50 μm. (C) Quantitative analysis of fipi and elff functional requirements for wing development. LOF conditions for fipi produced by the expression of UAS-CG15630 RNAi RRID: VDRC_107797kk (Ri1) or UAS-CG15630 RNAi RRID: BDSC_42589 (Ri2) under the control of the heterozygous MS1096-GAL4/+ driver caused a significant reduction of wing size (LZ as in Fig 2A). Over-expression of fipi (GOF1 and GOF2) in MS1096/+; UAS-fipi/+ individuals caused a stronger reduction in wing size. LOF conditions for elff produced by the expression of either UAS-CG33543 RNAi RRID: VDRC_32576 (Ri3) or UAS-CG33543 RNAi RRID: VDRC_40821 (Ri4) under the control of the MS1096-GAL4/+ heterozygous driver also caused a reduction of wing size. Over-expression of elff (GOF) in MS1096/+; UAS-elff/+ individuals also caused a reduction in wing size. The double LOF condition for fipi and elff (UAS-CG15630 RNAi RRID: BDSC_42589 plus UAS-CG33543 RNAi RRID: VDRC_32576, Ri2+3), produced a stronger wing size reduction, which was suppressed in a background with a 50% EGFR dosage reduction (Ri2+3 Egfr/+). All female individuals. Wing size is area in $\mu m^2/10^3$. (D) Quantification of activated-MAPK (ppERK) in ey-FLP driven FLP-OUTs. y ey-flp; Act5C-FRTy$^+$FRT-GAL4 UAS-GFP/CyO females were crossed to UAS-fipi$^{RNAi}$ and UAS-elff$^{RNAi}$ males, and to UAS-fas2$^{RNAi}$ (as an additional control). Larvae expressing the different RNAis were identified by their GFP expression in the eye disc, and sibling larvae devoid of GFP expression (CyO) were used as controls. Signal intensity (higher than 100, Image J Pixel Counter plugin, to avoid background signal noise) was quantified in the eye disc between the antenna disc and the morphogenetic furrow. In each case ppERK signal (MAPK signal) is presented as the ratio between experimental individuals and their control siblings. While the fas2 RNAi produces a strong decrease in MAPK activation with respect to the control siblings (see also [8]), the combined expression of fipi and elff RNAis produces a significant increase.

Expression of *fipi* or *elff RNAis* with a strong inhibiting capacity (see S3 Fig) (LOF) or their gene over-expression (GOF) under the control of the heterozygous *MS1096-GAL4/+* driver caused a significant reduction in wing size. The simultaneous expression of *fipi* and *elff RNAis* caused a strong phenotype, which was suppressed by decreasing *Egfr* dosage to 50% (Fig 5C). This indicates that the wing size reduction in these LOF combinations is consequence of a de-repression of the EGFR. Indeed, the double *fipi elff RNAi* combination raised at 29˚C caused the differentiation of extra-veins (a phenotype typical of *Egfr* GOF conditions), which were suppressed by reducing *Egfr* dosage (see below and Fig 6B, left panels). Moreover, comparison of activated-MAPK signal between the double *fipi elff RNAi* combination in *ey>FLP* driven FLP-OUTs with their normal sibling controls revealed a significant increase (Fig 5D). In accordance with an EGFR repressor function, over-expression of *UAS-fipi* caused loss of vein differentiation, an alteration typical of reduced *Egfr* function (Fig 6A, fipi$^{GOF}$ left bottom picture). Thus, the function of Fipi and Elff IgCAMs runs along that of Fas2 during imaginal disc growth, but they behave as EGFR repressors, opposite to Fas2 which functions as EGFR activator [8] (Fig 5D).

Expression of *fipi* or *elff RNAi* lines rescued the size of the wing in the *fas2* GOF condition (Fig 6A, top and middle pictures; Fig 6C), while *fipi* or *elff* GOF conditions reciprocally enhanced it (Fig 6A, bottom picture; Fig 6C). On the other hand, the combination of *fas2$^{GOF}$* with *fipi elff* RNAis displayed an enhanced wing size reduction and enhanced extra-vein territory differentiation (Fig 6B, top; C), which was suppressed by lowering *Egfr* dosage to 50% (Fig 6B, bottom; C). This result indicates that the Fas2 GOF can produce an increase in EGFR cell autonomous activity when released from the Fipi and Elff interaction, which synergizes with the *fipi elff* non-cell autonomous phenotype. Indeed, the analysis of activated-MAPK expression in combinations between the *fas2* GOF and *fipi* and *elff* RNAis revealed a significant increase over the MAPK activity level in the *fas2* GOF condition, for both the *fas2* GOF *elff* LOF and the *fas2* GOF *fipi elff* LOFs combinations (Fig 6D). To further confirm the increase of MAPK activity (ppERK) in *fipi elff* LOF and *fas2$^{GOF}$ fipi elff* LOF conditions, I used the reporter *Tub-miniCic* [27] in FLPOUT cell clones. Indeed, the *fipi elff* double RNAi LOF combination caused a decrease in the number of *miniCic*-positive nuclei, reflecting a non-cell

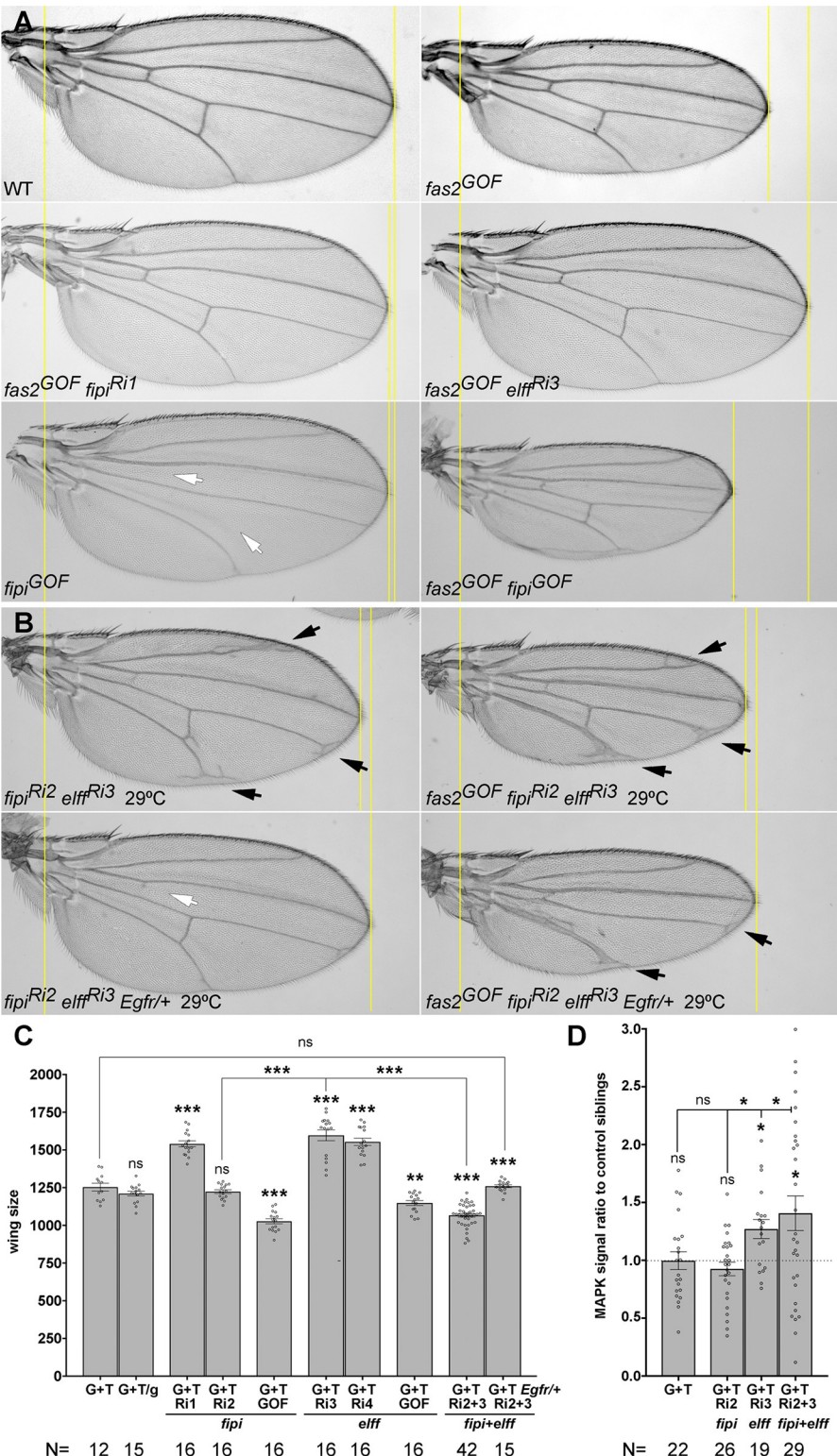

**Fig 6. The Fas2 non-cell autonomous function to repress growth is mediated by Fipi and Elff IgCAMs.** (A) The reduced wing size of the Fas2 GOF condition (fas2$^{GOF}$, MS1096-GAL4/+; UAS-fas$^{GPI}$ UAS-fas2$^{TRM}$/+; right top picture) is rescued by the LOF of either fipi$^{RNAi}$ (RRID:VDRC_107797kk, Ri1; left middle picture) or elff$^{RNAi}$ (RRID: VDRC_32576, Ri3; right middle picture). Over-expression of fipi under the control of heterozygous MS1096-GAL4/+ (fipi$^{GOF}$, bottom left picture) causes incomplete vein differentiation (white arrows) and strongly enhances the Fas2

GOF phenotype (fas2$^{GOF}$ fipi$^{GOF}$, bottom right picture). (B) The simultaneous inhibition of fipi and elff by RNAi (fipi$^{Ri2}$ elff$^{Ri3}$) expression driven by MS1096-GAL4/+ at 29°C causes a small wing and the differentiation of extra-veins (top left, black arrows), which is suppressed by reducing EGFR dosage by 50% (using Egfr$^{top}$, fipi$^{Ri2}$ elff$^{Ri3}$ Egfr/+, bottom left). At right, genetic interaction of the fas2 GOF condition with the double LOF for fipi and elff. The simultaneous inhibition of fipi and elff in the fas2 GOF condition displays enhanced growth loss and extra-vein formation (fas2$^{GOF}$ fipi$^{Ri2}$ elff$^{Ri3}$, top right picture), which is suppressed by a 50% EGFR dosage reduction (fas2$^{GOF}$ fipi$^{Ri2}$ elff$^{Ri3}$ Egfr/+, bottom right picture). (C) Quantitative analysis of the fas2$^{GOF}$ genetic interaction with fipi and elff. The fas2$^{GOF}$ (G+T, MS1096-GAL4/+; UAS-fas2$^{GPI}$ UAS-fas2$^{TRM}$/+, as in Fig 2A) wing phenotype is not modified by the introduction of an additional control UAS-GFP insertion (UAS-GFP, G+T/g). RNAi inhibition of fipi expression with fipi RNAi RRID: VDRC_107797kk (G+T Ri1) under the control of heterozygous MS1096-GAL4/+ rescues the fas2$^{GOF}$ wing phenotype, while over-expression of fipi (G+T GOF) causes a reciprocal enhancement. RNAi inhibition of elff expression using either elff RNAi RRID:VDRC_32576, (G+T Ri3) or elff RNAi RRID: VDRC_40821 (G+T Ri4) under the control of heterozygous MS1096-GAL4/+ rescues the fas2$^{GOF}$ wing phenotype, while over-expression of elff (G+T GOF) causes a reciprocal enhancement. The double fipi elff RNAi LOF (G+T Ri2+3) under the control of heterozygous MS1096-GAL4/+ causes an enhancement of the fas2$^{GOF}$ phenotype, which is rescued by reducing EGFR dosage to 50%. All female individuals. Wing size is area in µm$^2$/10$^3$. (D) Quantification of activated-MAPK (ppERK) in ey-FLP driven FLP-OUTs. y ey-flp; Act5C-FRTy$^+$FRT-GAL4 UAS-GFP/CyO females were crossed to UAS-fas2$^{GPI}$ UAS-fas2$^{TRM}$ (G+T), UAS-fas2$^{GPI}$ UAS-fas2$^{TRM}$ UAS-fipi$^{RNAi}$ (G+T Ri2), UAS-fas2$^{GPI}$ UAS-fas2$^{TRM}$ UAS-elff$^{RNAi}$ (G+T Ri3) and UAS-fas2$^{GPI}$ UAS-fas2$^{TRM}$ UAS-fipi$^{RNAi}$ UAS-elff$^{RNAi}$ (G+T Ri2+3) males. Larvae expressing the different RNAis were identified by their GFP expression in the eye disc, and sibling larvae devoid of GFP expression (CyO) were used as controls. Signal intensity (higher than 100, Image J Pixel Counter plugin, to avoid background signal noise) was quantified in the eye disc between the antenna disc and the morphogenetic furrow. In each case ppERK signal (MAPK signal) is presented as the ratio between experimental individuals and their control siblings. While the fas2$^{GOF}$ (G+T) shows a contained MAPK activation similar to control siblings, the combined expression of fas2$^{GOF}$ with fipi and elff RNAis produces a high increase.

autonomous increase in MAPK activity (Fig 7A and 7C). Moreover, the *fas2$^{GOF}$ fipi elff* LOF combination also showed a decrease in *miniCic*-positive nuclei compared to both sibling controls without clones and the *fas2* GOF condition (Fig 7A and 7D), again showing that the Fipi and Elff functions are epistatic over the Fas2 GOF phenotype. Thus, Fipi and Elff mediate the functional facet of Fas2 as non-cell autonomous repressor of EGFR, and in their absence the gain of EGFR activity can synergize with the Fas2 GOF functional facet as cell autonomous activator of EGFR.

The simultaneous Fas2 GOF and Fipi Elff LOF condition causes an adult phenotype similar to other EGFR over-activation conditions. Indeed, the extra-vein differentiation phenotype of the *fas2$^{GOF}$ fipi elff* LOF condition is strikingly similar to the extra-vein phenotype produced by *vein* over-expression (*UAS-vn*) (S4 Fig). Vn is a diffusible EGF, neuregulin-like, ligand activator of EGFR [28,29]. Interestingly, the *vn* over-expression phenotype was enhanced by the simultaneous expression of the *fas2* GOF (S4 Fig), suggesting that Vn and the cell autonomous function of Fas2 promoting EGFR activity can synergize during imaginal disc development.

To analyze the level of functional dependence of *fipi* and *elff* RNAis on *fas2* to produce the EGFR de-repression, I studied the combinations between them and a *fas2* RNAi (RRID: BDSC_28990). The results showed that *fipi* and *elff* RNAis can still produce an EGFR derepression phenotype in the absence of Fas2 (S5A and S5B Fig), albeit somewhat milder. The results are strongly consistent with Fipi and Elff being repressors of EGFR and enhancing their activity in interaction with a Fas2 high expression level (see Discussion).

To specifically analyze the interactions of Fas2 with Fipi and Elff at the end of larval growth, I introduced *Tub-GAL80ts* in the *fas2$^{GOF}$ fipi* and *elff* LOF combinations used above. Thus, allowing for the time-controlled expression of *fas2* GOF plus *fipi RNAi* or/and *elff RNAi* during the last half of third instar larval life. Both *fipi* and *elff* RNAis suppressed the *fas2* GOF, and the simultaneous inhibition of both *fipi* and *elff* produced now the expected stronger suppression of the *fas2* GOF (instead of the enhancement found when their expression was all along development) and an increase in vein territory (Fig 8A and 8B). Thus, Fipi and Elff behave as repressors of EGFR during imaginal disc development independent of Fas2, but by the end of

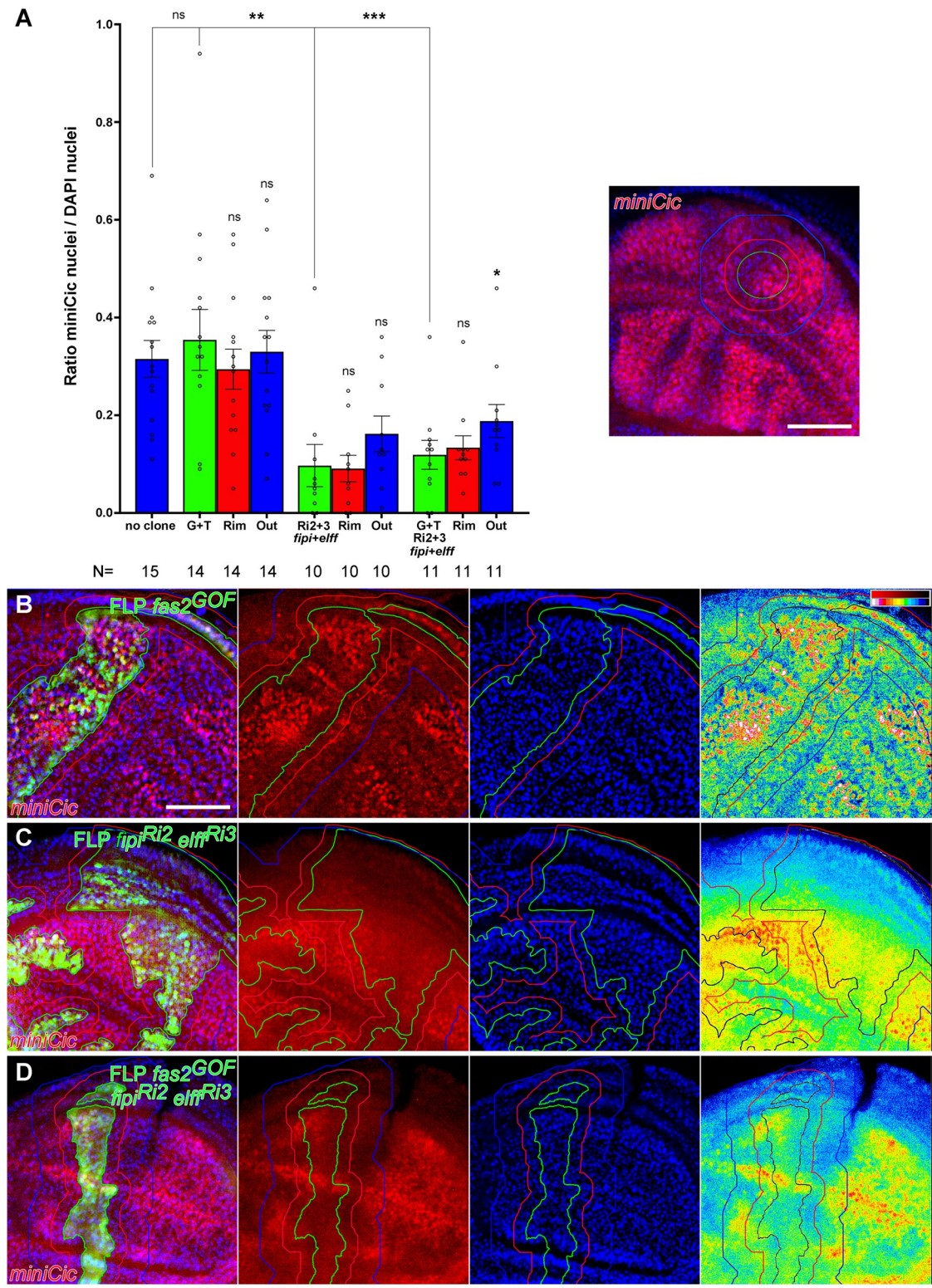

**Fig 7. Fipi and Elff are repressors of MAPK (ppERK) activity.** (A) Quantitative analysis of *Tub-miniCic-scarlett* nuclear expression in FLP-OUT clones of either Fas2 GOF (*UAS-fas2^GPI^ UAS-fas2^TRM^*; G+T), *UAS-fipi^RNAi^ UAS-elff*^RNAi^ (*CG15630 RNAi* RRID:BDSC_42589 and *CG33543 RNAi* RRID:VDRC_32576; Ri2+3 *fipi+elff*) or Fas2 GOF plus *UAS-fipi^RNAi^ UAS-elff*^RNAi^ (G+T Ri2+3 *fipi+elff*) in the wing disc. The MAPK (ppERK) activity reporter *miniCic* displays nuclear signal accumulation in conditions of low ERK activity, and depletion of nuclear signal in conditions of high ERK activity [25]. The Rim (red bars) corresponds to 2–3

normal cells neighbor to the clone, and Out (blue bars) corresponds to 4–9 normal cells from the clone. Y axis: fraction of DAPI nuclei showing *Tub-miniCic* signal. The inset shows the WT pattern of *miniCic* expression in late 3$^{rd}$ instar wing discs. Scale bar is 50 μm. (B) Fas2 GOF clones show a similar number of *miniCic*-positive nuclei to controls. The panel at right is a LUT of the red (*minCic*) signal, showing *miniCic* signal concentrated in many nuclei inside the clone. Clone territory outlined in green (black in the 16-color LUT), the rim of 2–3 normal cells around the clone in red and the halo of cells 4–9 cell diameters from the clone in blue. (C) FLPOUT clone expressing *fipi* and *elff* RNAis. Most cells have nuclei without *miniCic* signal which seems dispersed and more uniform than in B, indicating high MAPK activity. (D) *FLPOUT Fas2 GOF clone simultaneously expressing fipi and elff RNAis. The Fas2 GOF expression of nuclear miniCic is reduced (compare with B) and similar to the double fipi elff combination (compare to C), indicating high MAPK activity.*

imaginal disc growth both IgCAMs are mediating the non-cell autonomous function of Fas2 further restraining EGFR activity.

To further characterize the non-cell autonomous functional capacities of Fipi and Elff, I studied the mitotic index in Fas2 GOF FLP-OUT cell clones rescued by either *fipi* or *elff* RNAis. FLP-OUT clones over-expressing *fas2* along with *fipi RNAi* revealed a rescue of the mitotic index in the clones, the rims of 2–3 cells around them and in the more distant background epithelium (Fig 8C). In sharp contrast, a specific non-cell autonomous rescue of the mitotic index in the interface between Fas2-normal and Fas2-high cells (rim of 2–3 normal cells around the clone) occurred for the *fas2* GOF *elff RNAi* combination (Fig 8C). The results show that both Fipi and Elff IgCAMs are required in the same cell than Fas2 to produce its non-cell autonomous function, although Elff shows a shorter range of action than Fipi (Fig 8C and 8D). In summary, the results together suggest that a balance between Fas2-EGFR activation and Fipi/Elff-EGFR repression is required during imaginal disc growth, and this balance shifts towards a net EGFR repression once the level of expression of Fas2 has reached the threshold that allows for its interaction with Fipi and Elff (Fig 8E).

## Discussion

A general observation in the study of CAMs is that both low or increased levels of expression seem to reduce their performance. This prompted the principle that just the correct amount of adhesion is required for an optimal CAM-mediated interaction, with too little or too much stickiness affecting their performance [30,31]. In apparent accordance with this idea, Fas2 promotes maximal growth of synapses at an expression level of 50%, with lower or higher levels causing a smaller number of synaptic boutons [23]. This dependence on the Fas2 expression level directly translates into synaptic functionality [24]. On the other hand, for a signaling molecule this behavior would strongly suggest the presence of a negative feed-back starting at a given threshold. IgCAMs of the L1-CAM and NCAM families are not mere glues that support specific adhesion, they couple highly specific cell recognition and adhesion with the control of Receptor Tyrosine Kinase signaling [4–6]. Moreover, the function of Fas2 in cell signaling can be dissociated from its role in cell adhesion [14]. In addition, Fas2 can engage in protein-protein interactions with other IgCAMs through its extracellular domain [9]. Fas2 has been shown to promote EGFR function during axon extension [5] and imaginal disc cell proliferation [8], but has been reported to function as an EGFR repressor during *Drosophila* retinal differentiation [10]. Likewise, vertebrate NCAM has been shown to limit cell proliferation and promote differentiation in the nervous system [32], and be able to function as EGFR repressor [33]. Thus, Fas2 apparently behaves as either an EGFR activator or repressor depending on the developmental context. Our previous LOF analyses have revealed that Fas2 promotes EGFR function cell autonomously during imaginal disc growth and that, in turn, EGFR activity promotes Fas2 expression [8]. Interestingly, expression of Fas2 at the synapse has also been shown to depend on Ras activity [34], a main EGFR effector.

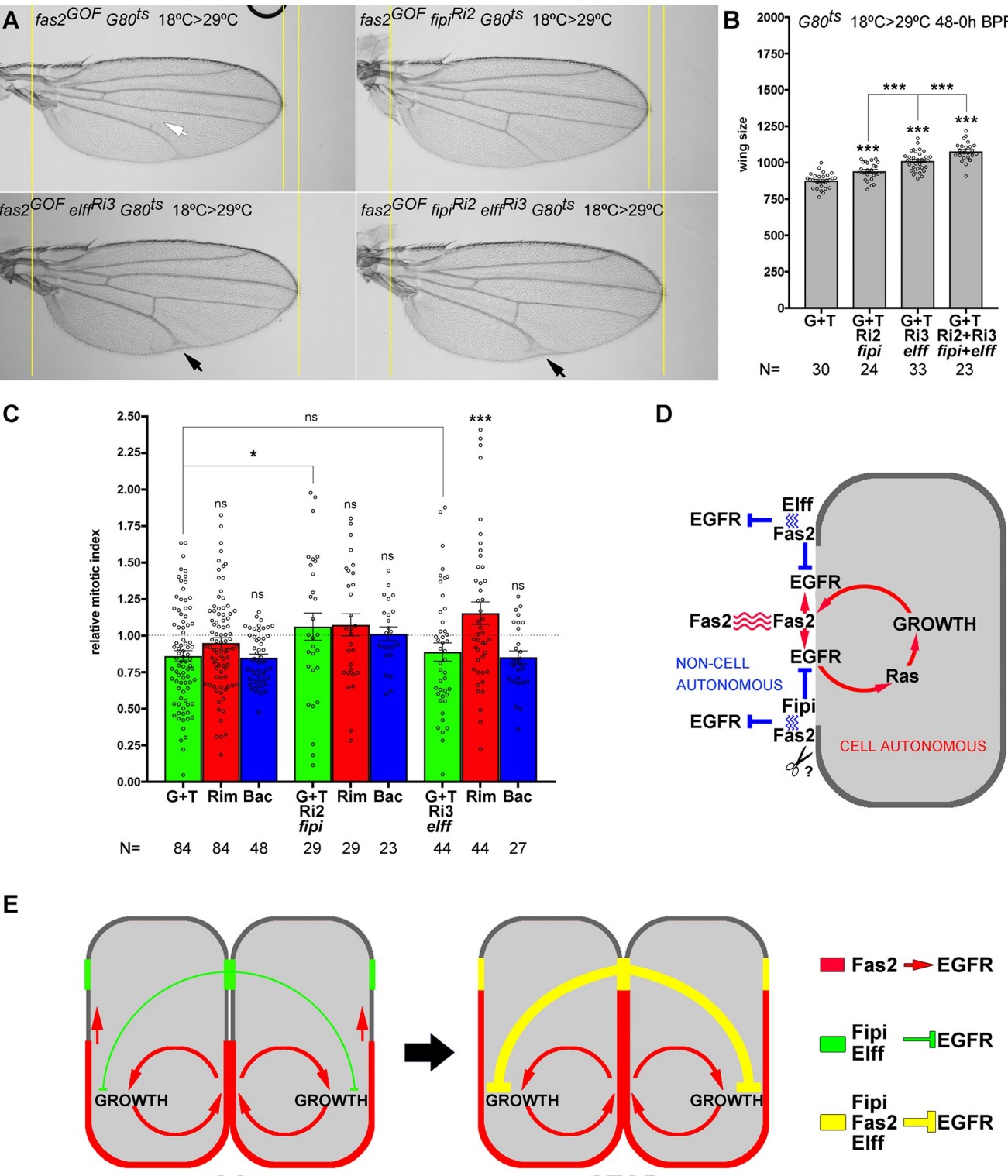

**Fig 8. Fas2 interacts with IgCAMs Fipi and Elff at the end of larval development to restrain growth and promote vein differentiation in the wing imaginal disc.** (A) Selective over-expression of Fas2 (fas2$^{GOF}$, MS1096-GAL4/+; TubG80$^{ts20}$/ UAS-fas2$^{GPI}$ UAS-fas2$^{TRM}$) in late 3$^{rd}$ instar larva causes the typical Fas2 GOF phenotype, including loss of cross-veins in some individuals (white arrow in top left picture). The selective inhibition of fipi (CG15630 RNAi RRID: BDSC_42589, Ri2) or elff (CG33543 RNAi RRID: VDRC_32576, Ri3) expression in the Fas2 GOF condition by the end of larval development suppresses its phenotype. Simultaneous inhibition of both fipi and elff in the Fas2 GOF condition at the end of larval development causes a stronger

suppression (bottom right picture) and an enhancement in the differentiation of extra-vein territory (black arrows, bottom pictures). (B) Quantification of the Fas2 GOF phenotype and its suppression by the selective inhibition of Fipi and Elff IgCAM expression at the end of larval life. MS1096-GAL4/+; TubG80$^{ts20}$/ UAS-fas$^{GPI}$ UAS-fas2$^{TRM}$ individuals display a stronger phenotypic suppression when the inhibition is simultaneous for fipi RNAi and elff RNAi (G+T Ri2+3). Larvae were shifted from 18˚C (permissive for GAL80ts) to 29˚C (restrictive for GAL80ts) during the second half of the third instar larva. All female individuals. Wing size is area in $\mu m^2/10^3$. (C) Quantitative analysis in FLP-OUT clones of the mitotic index (analyzed in late 3$^{rd}$ instar larva) in genetic interactions of fas2$^{GOF}$ with fipi and elff. Over-expression of Fas2 (UAS-fas2$^{GPI}$ plus UAS-fas2$^{TRM}$, G+T) in FLP-OUT cell clones causes a reduction of the mitotic index in the clone and in the rim of 2–3 normal neighbor cells around the clone (see also Fig 3B). Co-expression in these FLP-OUT clones of fipi RNAi (G+T Ri2) increases the mitotic index in the clone, the rim and farther out. Co-expression in the fas2$^{GOF}$ clones of elff RNAi (G+T Ri3) specifically rescues the non-cell autonomous effect in the rim of normal cells around the clone. Y axis: mitotic index relative to WT control FLP-OUT clones. (D) Summary of functional interactions found in the LOF [8] and GOF analyses of Fas2. Fas2 homophilic adhesion through its extracellular domains (red) is cell autonomously required to promote EGFR function by direct physical interaction. EGFR activity in turn promotes Fas2 expression creating a cell autonomous auto-stimulatory feedback loop. The Fas2 non-cell autonomous repression of EGFR requires the heterophilic interaction with Fipi and Elff in the same cell (in blue). Note that Fas2/Fipi may be released in the extracellular space (question mark) to mediate the Fas2 non-cell autonomous function beyond the first row of surrounding cells. (E) Model for a Fas2 expression-level switch mechanism controlling intercalary cell proliferation in imaginal discs. GO: growth in the proliferating epithelia of imaginal discs is maintained by a balance between the activation provided by the cell autonomous EGFR/Fas2 auto-stimulatory loop and the EGFR repression mediated by Fipi and Elff. STOP: during late larval development, increased Fas2 expression allows its interaction with Fipi and Elff tilting the EGFR activation/repression balance towards the restraining of growth.

In the present study, GOF analyses during imaginal disc development reveal that Fas2 can repress cell proliferation in a non-cell autonomous manner at high expression levels of the protein. This functional facet of Fas2 is mediated by the EGFR. Interestingly, Fas2 is expressed at peak levels in the morphogenetic furrow of the eye disc and in other pre-differentiating imaginal structures, like sensory organ precursor cells and veins in the wing disc [8]). Increasing the expression of Fas2 in imaginal disc clones early during larval development causes an EGFR-mediated reduction of clone size accompanied by a general reduction in final organ size. Remarkably, despite of this clone and organ decrease in size, the Fas2 GOF clones cover a larger fraction of the organ than normal control clones do, suggesting that the Fas2 GOF clones grow initially more than normal but the whole territory bearing these clones stop growth earlier than normal. Indeed, 3$^{rd}$ instar larva imaginal discs bearing these clones show a non-cell autonomous reduction of the mitotic index. Coupled-MARCM mosaics show that the Fas2 GOF repression of EGFR causes a decrease in the size of the control twins, and therefore it is remarkably different from the effect of altering the Fas2 homophilic interaction between cells in coupled-MARCM mosaics (which just affects the normal cells in direct contact with the clones [8]). Thus, the expression level of Fas2 can control cell proliferation in imaginal discs through the EGFR in both cell autonomous and non-cell autonomous opposite manners depending on its expression level. This raises the question if increased Fas2 expression during development could reach a threshold where its non-cell autonomous functional component could limit growth [8].

The results in this study show that the cell autonomous EGFR-activating and the non-cell autonomous EGFR-repressing functional facets of Fas2 seem to operate in a concerted manner during imaginal disc growth. While a strong inhibition of Fas2 expression during larval development impedes EGFR-dependent growth, a moderate decrease of its expression is sufficient to cause extra-growth (note the exact parallel with the situation of *fas2* requirement during synapse growth), consistent with an increased time to reach the above-mentioned critical threshold of Fas2 expression. Induction of the *fas2* mild LOF condition specifically at the end of larval growth confirms that Fas2 behaves as a repressor of growth by this stage. Thus, cells require the full normal amount of Fas2 to repress proliferation by the end of the growth period in imaginal discs.

The repression of EGFR function in the Fas2 GOF conditions is not a secondary effect of a putative high EGFR activity in the Fas2 GOF cells followed by the secretion of some feedback repressor, since a simple reduction in half of EGFR dosage or the expression of a mild DN-EGFR condition in the Fas2 GOF cells enhances the phenotype instead of suppressing it.

Rather, this shows Fas2 bearing a second functional facet at high expression level to actively repress EGFR in a non-cell autonomous manner. Therefore, Fas2 does not simply behave as a permissive factor for cell proliferation, but as an instructive one. The dissection of the cell autonomous and non-cell autonomous functional facets of Fas2 suggests the possible involvement of some heterophilic ligand on the cell surface which would mediate its non-cell autonomous function as EGFR repressor at high expression levels. Fipi and Elff IgCAMs have been shown to bind Fas2 through extracellular domain interactions [9]. Fipi and Elff are expressed just in apical apposition to Fas2 in the basolateral cell membrane of imaginal disc cells. Both proteins are required during imaginal disc growth, to repress vein formation in the wing disc and genetically behave as EGFR repressors. Moreover, their LOF conditions show increased MAPK activation and *miniCic* nuclear depletion, consistent with a derepression of EGFR. Their genetic interaction with *fas2* shows that both proteins mediate the Fas2 non-cell autonomous function. LOF conditions of *fipi* and *elff* suppress the Fas2 GOF phenotype, while their GOF conditions enhance it. Moreover, the specific inhibition of *fipi* or *elff* induced at the end of larval development, and specially their combined inhibition, suppresses the *fas2* GOF phenotype. Thus, by the end of larval development these IgCAMs restrict Fas2-dependent growth.

The genetic analysis of *fipi* and *elff* in combination with a *fas2* LOF condition shows that they can behave as repressors of EGFR on their own. The fact that the domains of expression of these IgCAMs in the membrane only partially overlap with Fas2 by the end of the larval stage is strongly consistent with the idea that protein-protein interactions between them may occur (or become more frequent) in high Fas2 expression situations. Thus, early RNAi inhibition of *fipi* and *elff* would cause an early derepression of EGFR but also the release of Fas2 over-expression from their EGFR-restraining interaction at the end of larval development, producing the strong derepression of EGFR observed in Fig 6B (with a phenotype similar to other EGFR de-repressed conditions). The analysis of both MAPK activation and *miniCic* nuclear depletion in the triple mutant ($fas2^{GOF}$ $fipi^{LOF}$ $elff^{LOF}$) combination confirms this inference. Therefore, it seems that during imaginal disc growth there is a balance between Fas2-dependent EGFR activation and Fipi/Elff-dependent EGFR repression, which tilts towards repression when Fas2 interacts with Fipi and Elff at the end of the larval stage (Fig 8E).

Although both Fipi and Elff are required in the same cell than the Fas2 GOF to mediate its non-cell autonomous EGFR repression, they seem to have a different way of action. Fipi is required in the Fas2 GOF cells to repress proliferation of both the Fas2 GOF cells and the normal territory beyond the first rows of contacting cells, suggesting that it may function by either shedding of the Fas2 extra-cytoplasmic domain as previously shown for NCAM [35] or mediated by cytonemes [36]. On the other hand, Elff is specifically required in the Fas2 GOF cells to repress proliferation of the closest normal cells around them, suggesting that it is mediated by direct cell to cell contact.

The integration of the Fas2, Fipi and Elff cell autonomous and non-cell autonomous functional facets may constitute a simple mechanism to drive and compute intercalary cell proliferation in the epithelium. The relative amount of these IgCAMs between cells would be critical to titrate the homophilic and heterophilic interaction balance towards promoting growth or restraining it. Amounts of Fas2 lower than a given threshold would favor the Fas2 homophilic interaction and promote EGFR/Ras pathway activity, producing in turn an increase of Fas2 expression due to its positive feedback with EGFR. Interestingly, NCAM-type proteins have been shown to require homophilic *cis* interactions for *trans* binding homophilic functional adhesion [37], and it has been proposed that they are required to build membrane molecular zippers [38]. Thus, Fas2 may be incorporated in the plasma membrane as a growing zipper during imaginal disc cell proliferation. This could progress until reaching over that Fas2-dependent species-specific expression threshold when heterophilic *cis* interactions with Fipi and

Elff would restrain EGFR activity and cell proliferation in a non-cell autonomous manner (Fig 8D and 8E). In addition, increasing Fas2 expression levels could synergize with diffusible EGFR ligands by coincident action, boosting EGFR activation and providing a high degree of cellular specificity to the action of these diffusible signals before imaginal disc differentiation. The interaction between the over-expression conditions of Vn and Fas2 confirms this inference. This Fas2-gain dependent growth model is reminiscent of the Entelechia model [39], but places IgCAM interactions as the key drivers behind the control of growth and shape. In this view, species-specific size and shape are the result of the interaction code and molecular affinities between IgCAMs on controlling RTKs activity.

During developmental growth and regeneration of epithelial organs, cell recognition is thought to drive intercalary cell proliferation. Cell recognition is also important to control axon and synapse growth and plasticity. It seems that Fas2 may also function as an expression-level switch in these processes as well. Fas2 and other CAMs that can engage in alternative homo- vs. heterophilic interactions are good candidates to constitute expression-level switches for the control of shape and size. CAM expression-level switches may constitute universal mechanisms to concert growth, size and shape control during organ development, axon extension and synapse formation.

## Experimental procedures

### Drosophila strains and genetic crosses

A description of the different *Drosophila* genes and mutations can be found at FlyBase, *www. flybase.org*. *Drosophila* stocks were obtained from the Bloomington Stock Center (BDSC) and Vienna Drosophila Research Center (VDRC). Plasmids containing full length cDNAs of *fipi* (*CG15630*) and *elff* (*CG33543*) were purchased from the Drosophila Genomics Resource Center, and the corresponding *UAS-fipi* and *UAS-elff* constructs and transformant lines were obtained from Best Gene, Inc., California. The *Tub-miniCic::scarlet* insertion was obtained from Romain Levayer. The different strains used in the mosaic analyses can be found in the Supplementary Materials and Methods. Timing for *Tub-GAL80ts* experiments was carried out in 24 hours egg collections and larvae were dated in hours before puparium formation (BPF). Clones were induced by 1-hour heat-shock at 37°C, for MARCM and coupled-MARCM analyses, and a 10 minutes heat-shock for FLP-OUT clones, during 1st (24–48 hours after egg laying, AEL) or 2nd (48–72 hours AEL) larval instars. For FLP-OUT clones in adult males, heat-shocks during 1st instar larva were of 5 minutes. All genetic crosses were maintained at 25 ±1°C in non-crowded conditions.

### Immunohistochemistry and data acquisition

Primary antibodies: rat anti-Elav for retinal neurons, anti-Fas2 MAb 1D4 and anti-βgal MAb 40-1a from the DSHB (University of Iowa); anti-βgal rabbit serum from Capel; rabbit anti-cleaved Caspase3 and anti-phospho-Histone H3B from Cell Signaling Technology; and ppERK (anti-MAPK) from Sigma. Staining protocols were standard according to antibody specifications. For the surface area measurements, images of adult nota and wings (one per individual) were acquired at 40X, pupal wing images at 100X, and imaginal discs at 200X or 400X. These images were used for surface measurements in pixels (pxs) and expressed as $\mu m^2$ according with the calibration of the microscope objective and digital camera used. Measured pupal wing area corresponded to the region from the alula to the tip of the wing after pupal molt (28–30 hours after puparium formation) and before wing expansion (40–44 hours after puparium formation). MAPK signal intensity (higher than 100, Image J Pixel Counter plugin, to avoid background signal noise) was quantified in the eye disc stained with anti-ppERK

between the antenna disc and the morphogenetic furrow. For colocalization analysis, I used ImageJ Colocalization and Colocalization Finder plugins (ratio: 50.0; threshold red: 100.0; threshold green: 100.0). Confocal images were obtained in a Leica DM-SL upright microscope (1024 x 1024 pxs camera), a Leica Thunder microscope (2048 x 2048 pxs camera), or in a Nikon Eclipse i80 microscope equipped with an Optigrid Structured Light System (1392 x 1040 pxs camera). Images were processed using Volocity 4.1–6.1 software, Improvision Ltd, Perkin-Elmer. I used ImageJ 1.52t LUT 16-colors in Fig 7B, the colocalization Plugin (threshold 100) in S2 Fig, and the color pixel counter plugin (threshold 100) for MAPK signal quantification.

## Statistical analysis

I measured clone area in pixels in Photoshop and transformed the values to $\mu m^2$ according with magnification and camera resolution. To calculate the mitotic index in the clone, its rim of normal cells and the rest of the imaginal disc, each GFP clone was outlined with the selection tool in Photoshop. To obtain the rim area of normal tissue corresponding to some 2–3 cell diameters around the GFP clone (camera 1024 x 1024), the outlined area was extended by 25 pixels out of the clone border and the area of the clone discarded. The area between this rim and the border of the disc defined the rest of the tissue. The mitotic index was calculated as the ratio of pH3 positive spots to the area in $\mu m^2$, and expressed as fractions of the corresponding value in the WT control clones. In the case of the FLPOUT *Tub-miniCic::scarlet* clones (camera 2048 x 2048 pxs), I counted the number of *miniCic*-positive nuclei and the total DAPI nuclei in the clone, the rim area of normal tissue corresponding to some 2–3 cell diameters around the GFP clone (50 pixels out of the clone border) and the halo (150 pxs out of the clone border). MAPK (ppERK) signal quantification was expressed as the ratio between the signal in the *ey-FLP* FLP-OUT imaginal discs and the mean in their CyO control siblings. I used paired Student's *t*-test (two-tailed) to compare the statistical differences between clone twins (in coupled-MARCM analysis) and between each clone, its rim and its imaginal disc background (in FLPOUT clones). I used unpaired Student's *t*-test (two-tailed) when variances were similar and fitted a Gaussian distribution, and Student's *t*-test with Welch's correction or the Mann-Whitney test when the variances were different or the distribution of values was not gaussian. Error bars in Figures are SEM. Significance value: * $P<0.05$, ** $P<0.01$ and *** $P<0.001$. Statistical software was Prism 4.0c and 7.0d, GraphPad Software, San Diego California USA, *www.graphpad.com*.

## Supporting information

**S1 Fig. Analysis of the Fas2 GOF condition in adult flies and FLP-OUT clones.** (A) Overexpression of Fas2$^{TRM}$ (*UAS-fas2$^{TRM}$*) under the control of the *Tub-GAL4/+* driver (*Tub-GAL4>fas2$^{GOF}$*) caused a general reduction of body size. Compare the two females, with overexpression of Fas2$^{TRM}$ (top) and a normal CyO sibling (*Tub-GAL4*) control (bottom). (B) A/P compartment border in an *en-GAL4 UAS-CD8GFP/UAS-fas2$^{TRM}$ UAS-fas2$^{GPI}$* pupal wing. The expression of Fas2$^{GPI}$ and Fas2$^{TRM}$ in the Posterior compartment of the wing (labelled with GFP) did not modify the size or shape of the cells. Bar: 50 $\mu m$. (C) FLP-OUT fas2 GOF clones in a puc-LacZ/+ genetic background did not show increased JNK activity nor apoptosis. At left, a wing disc with several Fas2 GOF clones (labeled with GFP) which cover most of the disc surface. The lack of ectopic expression from the reporter puc-LacZ reveals that the JNK pathway was not de-repressed in these clones. Staining with anti-cleaved Caspase 3 (Blue channel) did not reveal obvious signs of apoptosis in the wing discs either. Lower panels, note the presence of just three cells expressing cleaved Caspase 3 in one of the normal regions engulfed

by the Fas2 GOF clones. Bar: 50 μm. (D) FLP-OUT y w hs-FLP; ActinFRTy⁺FRT-GAL4 UAS-GFP/UAS-fas2^TRM^ UAS-fas2^GPI^; UAS-λEgfr/dpp-LacZ clones (middle panels) display the same phenotype than FLP-OUT y w hs-FLP; ActinFRTy⁺FRT-GAL4 UAS-GFP/+; UAS-λEgfr/dpp-LacZ clones (bottom panels). In both types of clones there is precocious non-cell autonomous retinal differentiation (arrows) and ectopic dpp expression inside the clone. Bar: 50 μm.
(TIF)

**S2 Fig. Fipi and Elff are expressed at the cell membrane and partially colocalize with Fas2 during late 3^rd^ instar larva.** (A) Top: the protein trap line *fipi-EGFP-FLAG* (RRID: BDSC_60532) shows protein colocalization with the membrane marker CD8-RFP. 3^rd^ instar wing imaginal disc stained with anti-Flag (green) and expressing *UAS-CD8-RFP* under the control of the *en-GAL4/+* driver. ImageJ Colocalization and Colocalization Finder plugins, ratio: 50.0; threshold red: 100.0; threshold green: 100.0; Pearson's correlation: + 0,349. Bottom: the expression of *fipi-EGFP-FLAG* (RRID:BDSC_60532) (anti-Flag, green) partially colocalizes with Fas2 (Mab 1D4, red) in the wing imaginal disc during late 3^rd^ instar larva. Expression of both proteins in different domains is better seen in the lateral views of cells at the imaginal disc folds. Note that the partial colocalization corresponds to the interface between Fas2 (red) and Fipi (green). ImageJ Colocalization and Colocalization Finder plugins, ratio: 50.0; threshold red: 100.0; threshold green: 100.0; Pearson's correlation: + 0,332. Bar: 50 μm. (B) Top: the protein trap line *elff-EGFP-FLAG* (RRID:BDSC_60531) also shows colocalization with the membrane marker CD8-RFP. 3^rd^ instar wing imaginal disc stained with anti-Flag (green) and expressing *UAS-CD8-RFP* under the control of the *en-GAL/+4* driver. ImageJ Colocalization and Colocalization Finder plugins, ratio: 50.0; threshold red: 100.0; threshold green: 100.0; Pearson's correlation: + 0,419. Bottom: *elff-EGFP-FLAG* (RRID:BDSC_60531) (anti-Flag, green) partially colocalizes with Fas2 (Mab 1D4, red) during late 3^rd^ instar larva. The panel shows the cell profiles at folds near the prospective wing pouch. Note the colocalization at the apposition of red (Fas2) and green (Elff) signals. ImageJ Colocalization and Colocalization Finder plugins, ratio: 50.0; threshold red: 100.0; threshold green: 100.0; Pearson's correlation: + 0,392.
(TIF)

**S3 Fig. *fipi* and *elff* RNAis efficiently suppress gene function.** The results for *fipi* and *elff* RNAi inhibition presented in Fig 5C are here compared with the same combinations over a deficiency that removes the corresponding endogenous gene. The phenotype caused by *UAS-fipi* or *UAS-elff* RNAi expression driven by MS1096/+ is not significantly enhanced by reducing 50% the dose of the corresponding gene using *Df(2L)BSC225, fipi⁻* (RRID:BDSC_9702) and *Df(2L)Exel7008, elff⁻* (RRID:BDSC_7780). The results show that these RNAis block most *fipi* and *elff* function causing at least a strong hypomorphic condition. Wing size is area in μm²/10³.
(TIF)

**S4 Fig. Over-expression of Fas2 enhances the EGFR activation caused by Vn over-expression.** (A) The gain of function for EGFR (*Egfr^GOF^*, MS1096-GAL4/+; *Egfr^λtop^*/+; bottom left) during imaginal wing disc growth produces adult wings smaller than normal (WT, *MS1096/+*, top left), and with a pronounced differentiation of extra-vein territory. A similar situation is attained by the over-expression of the EGF-like ligand Vein (*vn^GOF^*; top right). Both, the reduction in wing size and the extra-vein phenotype caused by the over- expression of *UAS-vn* (under the control of the *MS1096-GAL4/+* driver, *vn^GOF^*) is enhanced by the simultaneous over-expression of *fas2* (*vn^GOF^ fas2^GOF^*, MS1096/+; UAS-vn/UAS- fas2^GPI^ UAS-fas2^TRM^;

bottom right). (B) Quantification of wing area in over-expression conditions for Vn ($vn^{GOF}$) and Vn plus Fas2$^{GPI}$ Fas2$^{TRM}$ ($vn^{GOF}$ G+T) under the control of the *MS1096-GAL4/+* driver. All pictures heterozygous *MS1096-GAL4/+* females raised at 25˚C. Wing size is area in μm$^2$/10$^3$.
(TIF)

**S5 Fig. *fipi* and *elff* LOF conditions are epistatic to the *fas2* LOF.** (A) Inhibition of *fas2* (*RNAi* RRID: BDSC_28990) in combination with *fipi* (*RNAi* RRID:BDSC_42589, Ri2) or *elff* (*RNAi* RRID:VDRC_32576, Ri3) in *MS1096-GAL4/+* females causes a normalized adult wing size. The double inhibition of *fipi* and *elff* expression in the *fas2* LOF background shows a phenotype slightly suppressed. The combinations *fipi elff* are the same as in Fig 5C. All individuals are heterozygous *MS1096-GAL4/+* females raised at 25˚C. Wing size is area in μm$^2$/10$^3$. (B) The inhibition of *fas2* by the expression of *RNAi* (RRID:BDSC_28990) in *MS1096-GAL4/+* females (*fas2$^{LOF}$*) raised at 29˚C does not produce alterations in the wing vein pattern (with the exception of missing cross-veins in some individuals). Simultaneous inhibition of *fipi* and *elff* (*RNAi* RRID:BDSC_42589 and *RNAi* RRID:VDRC_32576) in *MS1096-GAL4/+* females (*fipi$^{Ri2}$ elff$^{Ri3}$*) raised at 29˚C causes the differentiation of extra-vein tissue (arrowheads), while the simultaneous inhibition of *fas2 fipi* and *elff* expression (*fipi$^{Ri2}$ elff$^{Ri3}$ fas2$^{LOF}$*) does not eliminate the formation of extra-veins (arrowheads). All pictures heterozygous *MS1096-GAL4/+* females raised at 29˚C.
(TIF)

**S1 File. Materials and methods.** Strains used for MARCM and coupled-MARCM analysis.
(DOCX)

**S2 File. Data and statistics.**
(ZIP)

# Acknowledgments

I am grateful to D. Ferres-Marco for thoughtful comments on the manuscript. I thank R. Levayer for providing the *Tub-miniCic-scarlet* insertion, and J. Morante and M. Dominguez for sharing resources. Most primary antibodies were obtained from the Developmental Studies Hybridoma Bank.

# Author Contributions

**Conceptualization:** Luis Garcia-Alonso.

**Data curation:** Luis Garcia-Alonso.

**Formal analysis:** Luis Garcia-Alonso.

**Investigation:** Luis Garcia-Alonso.

**Methodology:** Luis Garcia-Alonso.

**Project administration:** Luis Garcia-Alonso.

**Resources:** Luis Garcia-Alonso.

**Software:** Luis Garcia-Alonso.

**Supervision:** Luis Garcia-Alonso.

**Validation:** Luis Garcia-Alonso.

**Visualization:** Luis Garcia-Alonso.

**Writing – original draft:** Luis Garcia-Alonso.

**Writing – review & editing:** Luis Garcia-Alonso.

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
