## [Decision Letter · Decision Letter 0]

18 Jul 2024

PONE-D-24-18608Fasciclin 2 functions as an expression-level switch on EGFR to control organ shape and size in DrosophilaPLOS ONE

Dear Dr. Garcia-Alonso,

Thank you for submitting your manuscript to PLOS ONE. We have received comments from the reviewers. After careful consideration, we feel that manuscript has merit but minor revision is required. Therefore, we invite you to submit a revised version of the manuscript that addresses the points raised during the review process.

We look forward to receiving your revised manuscript.

Kind regards,

Ashutosh Pandey, Ph.D.

Academic Editor

PLOS ONE

 [This work was supported by PID2021-123407OB-I00, CEX2021-001165-S and BFU2016-76295-R from MCIN/AEI/10.13039/501100011033 and by “ERDF a way of making Europe”; PROMETEU 2021-027 from Generalitat Valenciana, and SAF2004-06593 from MCYT.].  

Please include this amended Role of Funder statement in your cover letter; we will change the online submission form on your behalf."

Additional Editor Comments (if provided):

Reviewers' comments:

Reviewer's Responses to Questions

**Comments to the Author**

1. Is the manuscript technically sound, and do the data support the conclusions?

Reviewer #1: Yes

Reviewer #2: Yes

2. Has the statistical analysis been performed appropriately and rigorously? 

Reviewer #1: Yes

Reviewer #2: Yes

3. Have the authors made all data underlying the findings in their manuscript fully available?

Reviewer #1: Yes

Reviewer #2: Yes

4. Is the manuscript presented in an intelligible fashion and written in standard English?

Reviewer #1: Yes

Reviewer #2: Yes

5. Review Comments to the Author

Reviewer #1: In this manuscript, the author has studied how Fasciclin 2, a cell adhesion molecule, controls organ size in fruit flies using wing disc as a model system. The author has used sophisticated genetic tools and techniques along with imaging studies to show a modulatory role of this gene on EGFR signaling. While the gain-of-function assays indicates a role of Fasciclin 2 as a non-cell autonomous repressor of EGFR, the loss-of-function assays indicate an interaction of this gene with other IgCAMs such as Fipi and Elff. It is interesting that depending on the specific cellular context, Fasciclin 2 can act cell autonomously as well as non-cell autonomously which may impact cell proliferation rate and organ size. Overall, this study is interesting and well done. Here are my suggestions:

1. The author may want to reorganize Fig1 and Fig.2 for better representation. Fig. 1A, B, D, F, I, and Fig. 2 A, C, F may be accommodated in Fig. 1. Rest should go to Fig. 2.

2. In Fig. 7C-D, the label written in green is not clearly legible. The author should use yellow or some other color so that the label is legible.

3. Also, do the author know the efficiency of the RNAi lines? Are all these lines published? If not, the author may want to test the efficiency of those lines.

4. Scale bar size should be written in the Figure legend for Fig. 1C.

5. Scale bar missing in Fig. 1G, 7B-D, S1B-D, and Fig. S2.

Reviewer #2: In this study, Fasciclin 2 (Fas2) emerges as a pivotal protein in the development of imaginal discs in Drosophila, which are precursors to adult structures like wings and the head capsule. Fas2 is shown to play a crucial role in normal development, but its overexpression causes significant effects on organ size and shape which is known from previous studies.

Overexpression of Fas2 results in a dose-dependent reduction in the size of adult organs, particularly wings and the head capsule. This overexpression also caused abnormal shapes in these organs, emphasizing the need for precise regulation of Fas2 levels during development.

Interestingly, the effects of Fas2 overexpression are non-cell autonomous, influencing neighbouring normal tissues as well. It also reduced their size without inducing cell death or activating common growth-regulating pathways like the JNK signaling pathway.

The data points out Fas2 overexpression represses the activity of EGFR (Epidermal Growth Factor Receptor) in neighbouring cells. This repression of EGFR is crucial for the observed reduction in organ size, indicating that Fas2's regulatory role involves interactions with EGFR signaling pathways.

Fipi and Elff, interact with Fas2 and are essential for its repressive function on EGFR at high expression levels. These interactions highlight the complexity of Fas2's regulatory network, where specific molecules like IgCAMs mediate its effects on EGFR signaling.

This manuscript provides insights into the multifaceted role of Fas2 in Drosophila development, particularly in regulating organ size and shape through its interactions with EGFR signaling pathways. The findings suggest a sophisticated regulatory network involving both cell-autonomous and non-cell autonomous mechanisms, mediated by specific protein interactions. Understanding these mechanisms not only sheds light on fundamental developmental processes but also underscores the importance of precise molecular regulation in ensuring proper tissue morphogenesis. It will be interesting to see how Fas2, and other CAMs engage in alternate homo or heteromeric interactions to regulate size and shape. Commend the authors on this decent study.

6. PLOS authors have the option to publish the peer review history of their article (what does this mean?). If published, this will include your full peer review and any attached files.

Reviewer #1: No

Reviewer #2: No

---

## [Author Response · Author response to Decision Letter 0]

11 Aug 2024

Additional Editor Comments (if provided): None

Reviewer #1: In this manuscript, the author has studied how Fasciclin 2, a cell adhesion molecule, controls organ size in fruit flies using wing disc as a model system. The author has used sophisticated genetic tools and techniques along with imaging studies to show a modulatory role of this gene on EGFR signaling. While the gain-of-function assays indicates a role of Fasciclin 2 as a non-cell autonomous repressor of EGFR, the loss-of-function assays indicate an interaction of this gene with other IgCAMs such as Fipi and Elff. It is interesting that depending on the specific cellular context, Fasciclin 2 can act cell autonomously as well as non-cell autonomously which may impact cell proliferation rate and organ size. Overall, this study is interesting and well done. Here are my suggestions:

1. The author may want to reorganize Fig1 and Fig.2 for better representation. Fig. 1A, B, D, F, I, and Fig. 2 A, C, F may be accommodated in Fig. 1. Rest should go to Fig. 2.

LGA response: Thank you very much for your suggestion. I changed Fig1 and Fig2 as follows: Fig2 A, B, C, E and F are now Fig1 C, D, J, K and L. This way all information for the wing disc is in Fig1. Fig1 C and G are now Fig2 A and B. Fig2 D is split now in Fig2 C and D. This way Fig2 presents all the information for the eye disc. I changed Figure legends correspondingly. I recognize that this way the information is better presented.

2. In Fig. 7C-D, the label written in green is not clearly legible. The author should use yellow or some other color so that the label is legible.

LGA response: I like that labels share the color of their subject. I have changed the position of the labels in the panels so they have a better visibility against the background.

3. Also, do the author know the efficiency of the RNAi lines? Are all these lines published? If not, the author may want to test the efficiency of those lines.

LGA response: I have added S3Fig showing the quantification of phenotype for the fipi and elff RNAis over a deficiency of the corresponding gene. As can be seen, the introduction of the deficiency does not significantly enhance the phenotype, showing that these RNAis are very efficient in suppressing most gene function from both endogenous copies of each gene.

4. Scale bar size should be written in the Figure legend for Fig. 1C.

LGA response: Fig 1C is now Fig2A. The scale is in the left picture at right.

5. Scale bar missing in Fig. 1G, 7B-D, S1B-D, and Fig. S2.

LGA response: Fig1G is now Fig2B, the scale in picture at right. Scale is now added in Fig7B. Scale is added now for S1FigB, C and D. Scale is now added in S2FigA.

Reviewer #2: In this study, Fasciclin 2 (Fas2) emerges as a pivotal protein in the development of imaginal discs in Drosophila, which are precursors to adult structures like wings and the head capsule. Fas2 is shown to play a crucial role in normal development, but its overexpression causes significant effects on organ size and shape which is known from previous studies.

Overexpression of Fas2 results in a dose-dependent reduction in the size of adult organs, particularly wings and the head capsule. This overexpression also caused abnormal shapes in these organs, emphasizing the need for precise regulation of Fas2 levels during development.

Interestingly, the effects of Fas2 overexpression are non-cell autonomous, influencing neighbouring normal tissues as well. It also reduced their size without inducing cell death or activating common growth-regulating pathways like the JNK signaling pathway.

The data points out Fas2 overexpression represses the activity of EGFR (Epidermal Growth Factor Receptor) in neighbouring cells. This repression of EGFR is crucial for the observed reduction in organ size, indicating that Fas2's regulatory role involves interactions with EGFR signaling pathways.

Fipi and Elff, interact with Fas2 and are essential for its repressive function on EGFR at high expression levels. These interactions highlight the complexity of Fas2's regulatory network, where specific molecules like IgCAMs mediate its effects on EGFR signaling.

This manuscript provides insights into the multifaceted role of Fas2 in Drosophila development, particularly in regulating organ size and shape through its interactions with EGFR signaling pathways. The findings suggest a sophisticated regulatory network involving both cell-autonomous and non-cell autonomous mechanisms, mediated by specific protein interactions. Understanding these mechanisms not only sheds light on fundamental developmental processes but also underscores the importance of precise molecular regulation in ensuring proper tissue morphogenesis. It will be interesting to see how Fas2, and other CAMs engage in alternate homo or heteromeric interactions to regulate size and shape. Commend the authors on this decent study.

LGA response: Thanks very much for your positive comments.

---

## [Editor Report · Decision Letter 1]

21 Aug 2024

Fasciclin 2 functions as an expression-level switch on EGFR to control organ shape and size in Drosophila

PONE-D-24-18608R1

Dear Dr. Garcia-Alonso,

We’re pleased to inform you that your manuscript has been reviewed scientifically suitable for publication and will be formally accepted for publication once it meets all outstanding technical requirements.

Kind regards,

Ashutosh Pandey, Ph.D.

Academic Editor

PLOS ONE
---

## [Editor Report · Acceptance letter]

10 Sep 2024

PONE-D-24-18608R1 

PLOS ONE

Dear Dr. Garcia-Alonso, 

I'm pleased to inform you that your manuscript has been deemed suitable for publication in PLOS ONE. Congratulations! Your manuscript is now being handed over to our production team.

Kind regards, 

on behalf of

Dr. Ashutosh Pandey 

Academic Editor

PLOS ONE